# Learning Self-Similarity in Space and Time as a Generalized Motion for Action Recognition

## Abstract

Spatio-temporal convolution often fails to learn motion dynamics in videos and thus an effective motion representation is required for video understanding in the wild. In this paper, we propose a rich and robust motion representation method based on *spatio-temporal self-similarity* (STSS). Given a sequence of frames, STSS represents each local region as similarities to its neighbors in space and time. By converting appearance features into relational values, it enables the learner to better recognize structural patterns in space and time. We leverage the whole volume of STSS and let our model learn to extract an effective motion representation from it. The proposed method is implemented as a neural block, dubbed *SELFY*, that can be easily inserted into neural architectures and learned end-to-end without additional supervision. With a sufficient volume of the neighborhood in space and time, it effectively captures long-term interaction and fast motion in the video, leading to robust action recognition. Our experimental analysis demonstrates its superiority over previous methods for motion modeling as well as its complementarity to spatio-temporal features from direct convolution. On the standard action recognition benchmarks, Something-Something-V1 & V2, Diving-48, and Fine-Gym, the proposed method achieves the state-of-the-art results.

## 1 Introduction

Learning spatio-temporal dynamics is the key to video understanding. To this end, extending convolutional neural networks (CNNs) with spatio-temporal convolution has been actively investigated in recent years (Tran et al., 2015; Carreira & Zisserman, 2017; Tran et al., 2018). The empirical results so far indicate that spatio-temporal convolution alone is not sufficient for grasping the whole picture; it often learns irrelevant context bias rather than motion information (Materzynska et al., 2020) and thus the additional use of optical flow turns out to boost the performance in most cases (Carreira & Zisserman, 2017; Lin et al., 2019). Motivated by this, recent action recognition methods learn to extract explicit motion, *i.e.*, flow or correspondence, between feature maps of adjacent frames and they improve the performance indeed (Li et al., 2020c; Kwon et al., 2020). But, is it essential to extract such an explicit form of flows or correspondences? How can we learn a richer and more robust form of motion information for videos in the wild?

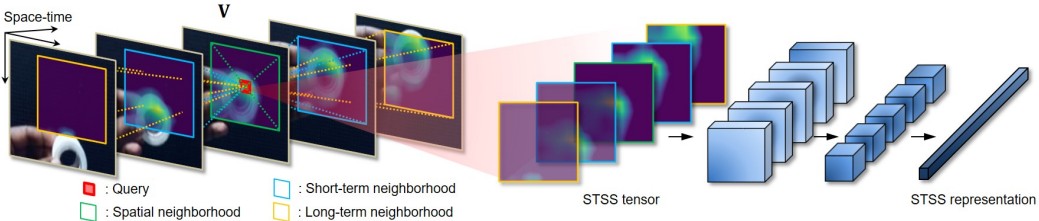

Figure 1: **Spatio-temporal self-similarity (STSS) representation learning**. STSS represents each spatio-temporal position (query) as its similarities (STSS tensor) with its neighbors in space and time (neighborhood). STSS allows to take a generalized, far-sighted view on motion, *i.e.*, both short-term and long-term, both forward and backward, as well as spatial self-motion. Our method learns to extract a rich and effective motion representation from STSS without additional supervision.

In this paper, we propose to learn *spatio-temporal self-similarity* (STSS) representation for video understanding. Self-similarity is a relational descriptor for an image that effectively captures intra-structures by representing each local region as similarities to its spatial neighbors (Shechtman & Irani, 2007). Given a sequence of frames, *i.e.*, a video, it extends along the temporal dimension and thus represents each local region as similarities to its neighbors in space and time. By converting appearance features into relational values, STSS enables a learner to better recognize structural patterns in space and time. For neighbors at the same frame it computes a spatial self-similarity map, while for neighbors at a different frame it extracts a motion likelihood map. If we fix our attention to the similarity map to the very next frame within STSS and attempt to extract a single displacement vector to the most likely position at the frame, the problem reduces to optical flow, which is a particular type of motion information. In contrast, we leverage the whole volume of STSS and let our model learn to extract an effective motion representation from it in an end-to-end manner. With a sufficient volume of neighborhood in space and time, it effectively captures long-term interaction and fast motion in the video, leading to robust action recognition.

We introduce a neural block for STSS representation, dubbed *SELFY*, that can be easily inserted into neural architectures and learned end-to-end without additional supervision. Our experimental analysis demonstrates its superiority over previous methods for motion modeling as well as its complementarity to spatio-temporal features from direct convolutions. On the standard benchmarks for action recognition, Something-Something V1&V2, Diving-48, and FineGym, the proposed method achieves the state-of-the-art results.

## 2 RELATED WORK

**Video action recognition**. Video action recognition is a task to categorize videos into pre-defined action classes. One of the conventional topics in action recognition is to capture temporal dynamics in videos. In deep learning, many approaches attempt to learn temporal dynamics in different ways: Two-stream networks with external optical flows (Simonyan & Zisserman, 2014; Wang et al., 2016), recurrent networks (Donahue et al., 2015), and 3D CNNs (Tran et al., 2015; Carreira & Zisserman, 2017). Recent approaches have introduced the advanced 3D CNNs (Tran et al., 2018; 2019; Feichtenhofer, 2020; Lin et al., 2019; Fan et al., 2020) and show the effectiveness of capturing spatio-temporal features, so that 3D CNNs now become a *de facto* approach to learn temporal dynamics in the video. However, spatio-temporal convolution is vulnerable unless relevant features are well aligned across frames within the fixed-sized kernel. To address this issue, a few methods adaptively translate the kernel offsets with deformable convolutions (Zhao et al., 2018; Li et al., 2020a), while several methods (Feichtenhofer et al., 2019; Li et al., 2020b) modulate the other hyper-parameters, *e.g.*, higher frame rate or larger spatial receptive fields. Unlike these methods, we address the problem of the spatio-temporal convolution by a sufficient volume of STSS, capturing far-sighted spatio-temporal relations.

**Learning motion features.** Since using the external optical flow benefits 3D CNNs to improve the action recognition accuracy (Carreira & Zisserman, 2017; Zolfaghari et al., 2018; Tran et al., 2018), several approaches try to learn frame-by-frame motion features from RGB sequences inside neural architectures. Fan et al. (2018); Piergiovanni & Ryoo (2019) internalize TV-L1 (Zach et al., 2007) optical flows into the CNN. Frame-wise feature differences (Sun et al., 2018b; Lee et al., 2018; Jiang et al., 2019; Li et al., 2020c) are also utilized as the motion features. Recent correlation-based methods (Wang et al., 2020; Kwon et al., 2020) adopt a correlation operator (Dosovitskiy et al., 2015; Sun et al., 2018a; Yang & Ramanan, 2019) to learn motion features between adjacent frames. However, these methods compute frame-by-frame motion features between two adjacent frames and then rely on stacked spatio-temporal convolutions for capturing long-range motion dynamics. We propose to learn STSS features, as generalized motion features, that enable to capture both short-term and long-term interactions in the video.

**Self-similarity.** Self-similarity represents an internal geometric layout of images. It is widely used in many computer vision tasks, such as object detection (Shechtman & Irani, 2007), image retrieval (Hörster & Lienhart, 2008), and semantic correspondence matching (Kim et al., 2015; 2017). In the video domain, Shechtman & Irani (2007) firstly introduce the concept of STSS and transforms the STSS to a hand-crafted local descriptor for action detection. Inspired from this work, early methods adopt self-similarities for capturing view-invariant temporal patterns (Junejo et al.,

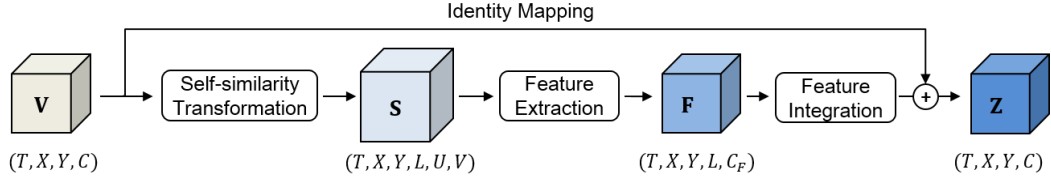

Figure 2: **Overview of our self-similarity representation block (SELFY)**. SELFY block takes as input a video feature tensor $\mathbf{V}$, transforms it to a STSS tensor $\mathbf{S}$, and extracts a feature tensor $\mathbf{F}$ from $\mathbf{S}$. It then converts $\mathbf{F}$ to the same size as the input $\mathbf{V}$ via the feature integration, and combines it with the input $\mathbf{V}$ to produce the final STSS representation $\mathbf{Z}$. See text for details.

2008; 2010; Körner & Denzler, 2013), but they use temporal self-similarities only due to computational costs. Recently, there are several non-local approaches (Wang et al., 2018; Liu et al., 2019) that utilize STSS for capturing long-range dynamics of videos. However, they use STSS for re-weighting or aligning visual features, which is an indirect way of using STSS. Different from these methods, our method leverages full STSS directly as generalized motion information and learns an effective representation for action recognition within a video-processing architecture. To the best of our knowledge, our work is the first in learning STSS representation using modern CNNs.

The contribution of our paper can be summarized as follows. First, we revisit the notion of self-similarity and propose to learn a generalized, far-sighted motion representations from STSS. Second, we implement STSS representation learning as a neural block, dubbed *SELFY*, that can be integrated into existing neural architectures. Third, we provide comprehensive evaluations on SELFY, achieving the state-of-the-art on benchmarks: Something-Something V1&V2, Diving-48, and FineGym.

## 3 OUR APPROACH

In this section, we first revisit the notions of self-similarity and discuss its relation to motion. We then introduce our method for learning effective spatio-temporal self-similarity representation, which can be easily integrated into video-processing architectures and learned end-to-end.

### 3.1 SELF-SIMILARITY REVISITED

Self-similarity is a relational descriptor that suppresses variations in appearance and reveals structural patterns in images or videos (Shechtman & Irani, 2007).

Given an image feature map $\mathbf{I} \in \mathbb{R}^{X \times Y \times C}$, self-similarity transformation of $\mathbf{I}$ results in a 4D tensor $\mathbf{S} \in \mathbb{R}^{X \times Y \times U \times V}$, whose elements are defined as

$$\mathbf{S}_{x,y,u,v} = \text{sim}(\mathbf{I}_{x,y}, \mathbf{I}_{x+u,y+v}),$$

where $\text{sim}(\cdot, \cdot)$ is a similarity function, *e.g.*, cosine similarity. Here, $(x, y)$ is a query coordinate while $(u, v)$ is a spatial offset from it. To impose a locality, the offset is restricted to its neighborhood: $(u, v) \in [-d_\text{U}, d_\text{U}] \times [-d_\text{V}, d_\text{V}]$, so that $U = 2d_\text{U} + 1$ and $V = 2d_\text{V} + 1$, respectively. By converting $C$-dimensional appearance feature $\mathbf{I}_{x,y}$ into $UV$-dimensional relational feature $\mathbf{S}_{x,y}$, it suppresses variations in appearance and reveals spatial structures in the image. Note that the self-similarity transformation closely relates to conventional cross-similarity (or correlation) across two different feature maps ($\mathbf{I}, \mathbf{I}' \in \mathbb{R}^{X \times Y \times C}$), which can be defined as

$$\mathbf{S}_{x,y,u,v} = \text{sim}(\mathbf{I}_{x,y}, \mathbf{I}'_{x+u,y+v}).$$

Given two images of a moving object, the cross-similarity transformation effectively captures motion information and thus is commonly used in optical flow and correspondence estimation (Dosovitskiy et al., 2015; Sun et al., 2018a; Yang & Ramanan, 2019).

For a sequence of frames, *i.e.*, a video, one can naturally extend the spatial self-similarity along the temporal axis. Let $\mathbf{V} \in \mathbb{R}^{T \times X \times Y \times C}$ denote a feature map of the video with $T$ frames. *Spatio-temporal self-similarity* (STSS) transformation of $\mathbf{V}$ results in a 6D tensor $\mathbf{S} \in \mathbb{R}^{T \times X \times Y \times L \times U \times V}$, whose elements are defined as

$$\mathbf{S}_{t,x,y,l,u,v} = \text{sim}(\mathbf{V}_{t,x,y}, \mathbf{V}_{t+l,x+u,y+v}), \tag{1}$$

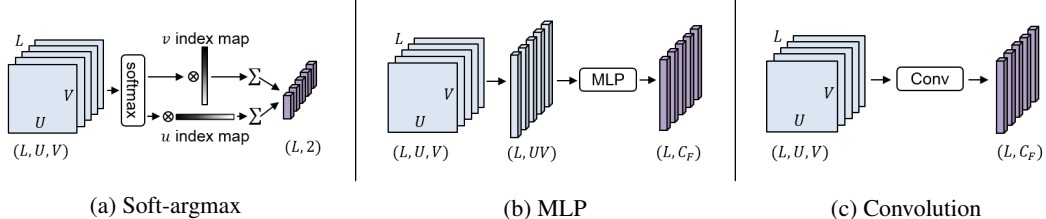

(a) Soft-argmax      (b) MLP      (c) Convolution

Figure 3: **Feature extraction from STSS.** For a spatio-temporal position $(t, x, y)$, each method transforms $(L, U, V)$ volume of STSS tensor $\mathbf{S}$ into $(L, C_F)$. See text for details.

where $(t, x, y)$ is the spatio-temporal coordinate and $(l, u, v)$ is a spatio-temporal offset from it. In addition to the locality of spatial offsets above, the temporal offset $l$ is also restricted to its temporal neighborhood: $l \in [-d_{\mathrm{L}}, d_{\mathrm{L}}]$, so that $L = 2d_{\mathrm{L}} + 1$.

What types of information does STSS describe? Interestingly, for each time $t$, the STSS tensor $\mathbf{S}$ can be decomposed along temporal offset $l$ into a single spatial self-similarity tensor (when $l = 0$) and $2d_{\mathrm{L}}$ spatial cross-similarity tensors (when $l \neq 0$); the partial tensors with a small offset (*e.g.*, $l = -1$ or $+1$) collect motion information from adjacent frames and those with larger offsets capture it from further frames both forward and backward in time. Unlike previous approaches to learn motion (Dosovitskiy et al., 2015; Wang et al., 2020; Kwon et al., 2020), which rely on cross-similarity between adjacent frames, STSS allows to take a generalized, far-sighted view on motion, *i.e.*, both short-term and long-term, both forward and backward, as well as spatial self-motion.

## 3.2 Spatio-temporal self-similarity representation learning

By leveraging the rich information in STSS, we propose to learn a generalized motion representation for video understanding. To achieve this goal without additional supervision, we design a neural block, dubbed SELFY, which can be inserted into a video-processing architectures and learned end-to-end. The overall structure is illustrated in Fig. 2. It consists of three steps: *self-similarity transformation*, *feature extraction*, and *feature integration*.

Given the input video feature tensor $\mathbf{V}$, the self-similarity transformation step converts it to the STSS tensor $\mathbf{S}$ as in Eq.(1). In the following, we describe feature extraction and integration steps.

### 3.2.1 Feature extraction

From the STSS tensor $\mathbf{S} \in \mathbb{R}^{T \times X \times Y \times L \times U \times V}$, we extract a $C_F$-dimensional feature for each spatio-temporal position $(t, x, y)$ and temporal offset $l$ so that the resultant tensor is $\mathbf{F} \in \mathbb{R}^{T \times X \times Y \times L \times C_F}$, which is equivariant to translation in space, time, and temporal offset. The dimension of $L$ is preserved to extract motion information across different temporal offsets in a consistent manner. While there exist many design choices, we introduce three methods for feature extraction in this work.

**Soft-argmax.** The first method is to compute explicit displacement fields using $\mathbf{S}$, which previous motion learning methods adopt using spatial cross-similarity (Dosovitskiy et al., 2015; Sun et al., 2018a; Yang & Ramanan, 2019). One may extract the displacement field by indexing the positions with the highest similarity value via $\arg\max_{(u,v)}$, but it is not differentiable. We instead use *soft-argmax* (Chapelle & Wu, 2010), which aggregates displacement vectors with softmax weighting (Fig. 3a). The soft-argmax feature extraction can be formulated as

$$\mathbf{F}_{t,x,y,l} = \sum_{u,v} \frac{\exp(\mathbf{S}_{t,x,y,l,u,v}/\tau)}{\sum_{u',v'} \exp(\mathbf{S}_{t,x,y,l,u',v'}/\tau)} [u; v], \tag{2}$$

which results in a feature tensor $\mathbf{F} \in \mathbb{R}^{T \times X \times Y \times L \times 2}$. The temperature factor $\tau$ adjusts the softmax distribution, and we set $\tau = 0.01$ in our experiments.

**Multi-layer perceptron (MLP).** The second method is to learn an MLP that converts self-similarity values into a feature. For this, we flatten the $(U, V)$ volume into $UV$-dimensional vectors, and apply an MLP to them (Fig. 3b). For the reshaped tensor $\mathbf{S} \in \mathbb{R}^{T \times X \times Y \times L \times UV}$, a perceptron $f(\cdot)$ can be expressed as

$$f(\mathbf{S}) = \mathrm{ReLU}(\mathbf{S} \times_5 \mathbf{W}_\phi), \tag{3}$$

where $\times_n$ denotes the $n$-mode tensor product, $\mathbf{W}_\phi \in \mathbb{R}^{C' \times UV}$ is the perceptron parameters, and the output is $f(\mathbf{S}) \in \mathbb{R}^{T \times X \times Y \times L \times C'}$. The MLP feature extraction can thus be formulated as

$$\mathbf{F} = (f_n \circ f_{n-1} \circ \cdots \circ f_1)(\mathbf{S}), \tag{4}$$

which produces a feature tensor $\mathbf{F} \in \mathbb{R}^{T \times X \times Y \times L \times C_F}$. This method is more flexible and may also be more effective than the soft-argmax because not only can it encode displacement information but also it can directly access the similarity values, which may be helpful for learning motion distribution.

**Convolution.** The third method is to learn convolution kernels over $(L, U, V)$ volume of $\mathbf{S}$ (Fig. 3c). When we regard $\mathbf{S}$ as a 7D tensor $\mathbf{S} \in \mathbb{R}^{T \times X \times Y \times L \times U \times V \times C}$ with $C = 1$, the convolution layer $g$ can be expressed using $\mathbf{S}' = g(\mathbf{S})$, whose elements are computed by

$$\mathbf{S}'_{t,x,y,l,u,v,c'} = \mathrm{ReLU}\Big( \sum_{l_\kappa, u_\kappa, v_\kappa, c} \mathbf{K}_{l_\kappa, u_\kappa, v_\kappa, c, c'} \mathbf{S}_{t,x,y,l+\hat{l}_\kappa, u+\hat{u}_\kappa, v+\hat{v}_\kappa, c} \Big). \tag{5}$$

where $\mathbf{K} \in \mathbb{R}^{L_\kappa \times U_\kappa \times V_\kappa \times C \times C'}$ is a multi-channel convolution kernel, $(l_\kappa, u_\kappa, v_\kappa)$ is the kernel parameter indices, and $(c, c')$ is the channel indices. The indices $(\hat{l}_\kappa, \hat{u}_\kappa, \hat{v}_\kappa)$ are centered as $\hat{l}_\kappa = l_\kappa - L_\kappa/2, \hat{u}_\kappa = u_\kappa - U_\kappa/2, \hat{v}_\kappa = v_\kappa - V_\kappa/2$. Starting from $\mathbb{R}^{T \times X \times Y \times L \times U \times V \times 1}$, we gradually downsample (U,V) and expand channels via multiple convolutions with strides, finally resulting in $\mathbb{R}^{T \times X \times Y \times L \times 1 \times 1 \times C_F}$; we preserve the L dimension, since maintaining fine temporal resolution is shown to be effective for capturing detailed motion information (Lin et al., 2019; Feichtenhofer et al., 2019). The convolutional feature extraction with $n$ layers can thus be formulated as

$$\mathbf{F} = (g_n \circ g_{n-1} \circ \cdots \circ g_1)(\mathbf{S}), \tag{6}$$

which results in a feature tensor $\mathbf{F} \in \mathbb{R}^{T \times X \times Y \times L \times C_F}$. This method is effective in learning structural patterns with their convolution kernels, thus outperforming the former methods as will be seen in our experiments.

### 3.2.2 Feature integration

In this step, we integrate the extracted STSS features $\mathbf{F} \in \mathbb{R}^{T \times X \times Y \times L \times C_F}$ to feed them back to the original input stream with $(T, X, Y, C)$ volume.

We first use $3 \times 3$ spatial convolution kernels along $(x, y)$ dimension of $\mathbf{F}$. The spatial convolution layer $h$ can be expressed using $\mathbf{F}' = h(\mathbf{F})$, whose elements are computed by

$$\mathbf{F}'_{l,x,y,t,c'_F} = \mathrm{ReLU}\Big( \sum_{(x_\kappa, y_\kappa, c'_F)} \mathbf{K}_{x_\kappa, y_\kappa, c_F, c'_F} \mathbf{F}_{l, x+\hat{x}_\kappa, y+\hat{y}_\kappa, t, c_F} \Big), \tag{7}$$

where $\mathbf{K} \in \mathbb{R}^{X_\kappa \times Y_\kappa \times C_F \times C'_F}$ is the multi-channel convolution kernel, $(x_\kappa, y_\kappa)$ is the kernel parameter indices, and $(c_F, c'_F)$ is the channel indices. $(\hat{x}_\kappa, \hat{y}_\kappa)$ is centered as $\hat{x}_\kappa = x_\kappa - X_\kappa/2, \hat{y}_\kappa = y_\kappa - Y_\kappa/2$. This type of spatial convolutions integrate the original features by extending receptive fields along $(x, y)$ dimension. The resultant features $\mathbf{F}^\star \in \mathbb{R}^{T \times X \times Y \times L \times C_F^\star}$ is defined as

$$\mathbf{F}^\star = (h_n \circ h_{n-1} \circ \cdots \circ h_1)(\mathbf{F}). \tag{8}$$

We then flatten the $(L, C_F^\star)$ volume into $LC_F^\star$-dimensional vectors to obtain $\mathbf{F}^\star \in \mathbb{R}^{T \times X \times Y \times LC_F^\star}$, and apply an $1 \times 1 \times 1$ convolution layer to obtain the final output. This convolution layer integrates features from different temporal offsets and also adjusts its channel dimension to fit that of the original input $\mathbf{V}$. We adopt the identity mapping of the input for residual learning (He et al., 2016). The final output tensor $\mathbf{Z}$ is expressed as

$$\mathbf{Z} = \mathrm{ReLU}(\mathbf{F}^\star \times_5 \mathbf{W}_\theta) + \mathbf{V}, \tag{9}$$

where $\times_n$ is the $n$-mode tensor product and $\mathbf{W}_\theta \in \mathbb{R}^{C \times LC_F^\star}$ is the weights of the convolution layer.

## 4 Experiments

### 4.1 Datasets & Implementation details

For evaluation, we use benchmarks that contain fine-grained spatio-temporal dynamics in videos.

Table 1: **Performance comparison on SS-V1&V2**. Top-1, 5 accuracy (%) and FLOPs (G) are shown.

| model | flow | #frame | FLOPs ×clips | SS-V1 top-1 | SS-V1 top-5 | SS-V2 top-1 | SS-V2 top-5 |
|---|---|---|---|---|---|---|---|
| TSN-R50 from (Lin et al., 2019) | | 8 | 33 G×1 | 19.7 | 46.6 | 30.0 | 60.5 |
| TRN-BNIncep (Zhou et al., 2018) | | 8 | 16 G×N/A | 34.4 | - | 48.8 | - |
| TRN-BNIncep Two-stream (Zhou et al., 2018) | ✓ | 8+8 | 16 G×N/A | 42.0 | - | 55.5 | - |
| MFNet-R50 (Lee et al., 2018) | | 10 | N/A×10 | 40.3 | 70.9 | - | - |
| CPNet-R34 (Liu et al., 2019) | | 24 | N/A×96 | - | - | 57.7 | 84.0 |
| TPN-R50 (Yang et al., 2020) | | 8 | N/A×10 | 40.6 | - | 59.1 | - |
| SELFYNet-R50 (ours) | | 8 | 37 G×1 | **50.7** | **79.3** | **62.7** | **88.0** |
| I3D from (Wang & Gupta, 2018) | | 32 | 153 G×2 | 41.6 | 72.2 | - | - |
| NL-I3D from (Wang & Gupta, 2018) | | 32 | 168 G×2 | 44.4 | 76.0 | - | - |
| TSM-R50 (Lin et al., 2019) | | 16 | 65 G×1 | 47.3 | 77.1 | 61.2 | 86.9 |
| TSM-R50 Two-stream from (Kwon et al., 2020) | ✓ | 16+16 | 129 G×1 | 52.6 | 81.9 | 65.0 | 89.4 |
| CorrNet-R101 (Wang et al., 2020) | | 32 | 187 G×10 | 50.9 | - | - | - |
| STM-R50 (Jiang et al., 2019) | | 16 | 67 G×30 | 50.7 | 80.4 | 64.2 | 89.8 |
| TEA-R50 (Li et al., 2020c) | | 16 | 70 G×30 | 52.3 | 81.9 | - | - |
| MSNet-TSM-R50 (Kwon et al., 2020) | | 16 | 67 G×1 | 52.1 | 82.3 | 64.7 | 89.4 |
| MSNet-TSM-R50$_{EN}$ (Kwon et al., 2020) | | 16+8 | 101 G×10 | 55.1 | 84.0 | 67.1 | 91.0 |
| SELFYNet-TSM-R50 (ours) | | 8 | 37 G×1 | 52.5 | 80.8 | 64.5 | 89.4 |
| SELFYNet-TSM-R50 (ours) | | 16 | 77 G×1 | 54.3 | 82.9 | 65.7 | 89.8 |
| SELFYNet-TSM-R50$_{EN}$ (ours) | | 8+16 | 114 G×1 | 55.8 | 83.9 | 67.4 | 91.0 |
| SELFYNet-TSM-R50$_{EN}$ (ours) | | 8+16 | 114 G×2 | **56.6** | **84.4** | **67.7** | **91.1** |

Table 2: **Performance comparison on Diving-48 & FineGym.**

(a) **Performance comparison on Diving-48**. Top-1 accuracy (%) and FLOPs (G) are shown.

| model | #frame | FLOPs ×clips | Top-1 |
|---|---|---|---|
| TSN from (Li et al., 2018) | - | - | 16.8 |
| TRN from (Kanojia et al., 2019) | - | - | 22.8 |
| Att-LSTM (Kanojia et al., 2019) | 64 | N/A×1 | 35.6 |
| P3D from (Luo & Yuille, 2019) | 16 | N/A×1 | 32.4 |
| C3D from (Luo & Yuille, 2019) | 16 | N/A×1 | 34.5 |
| GST-R50 (Luo & Yuille, 2019) | 16 | 59 G×1 | 38.8 |
| CorrNet-R101 (Wang et al., 2020) | 32 | 187 G×10 | 38.2 |
| GSM-IncV3 (Sudhakaran et al., 2020) | 16 | 54 G×2 | 40.3 |
| TSM-R50 (our impl.) | 16 | 65 G×2 | 38.8 |
| SELFYNet-TSM-R50 (ours) | 16 | 77 G×2 | **41.6** |

(b) **Performance comparison on FineGym**. The averaged per-class accuracy (%) is shown. All results in the upper part are from Shao et al. (2020).

| model | #frame | Gym288 Mean | Gym99 Mean |
|---|---|---|---|
| TSN (Wang et al., 2016) | 3 | 26.5 | 61.4 |
| TRN (Zhou et al., 2018) | 3 | 33.1 | 68.7 |
| I3D (Carreira & Zisserman, 2017) | 8 | 27.9 | 63.2 |
| NL I3D (Wang et al., 2018) | 8 | 27.1 | 62.1 |
| TSM (Lin et al., 2019) | 3 | 34.8 | 70.6 |
| TSM Two-stream (Lin et al., 2019) | N/A | 46.5 | 81.2 |
| TSM-R50 (our impl.) | 3 | 35.3 | 73.7 |
| TSM-R50 (our impl.) | 8 | 47.9 | 86.6 |
| SELFYNet-TSM-R50 (ours) | 8 | **49.5** | **87.7** |

**Something-Something V1 & V2 (SS-V1 & V2)** (Goyal et al., 2017b), which are both large-scale action recognition datasets, contain ∼108k and ∼220k video clips, respectively. Both datasets share the same 174 action classes that are labeled, *e.g.*, 'pretending to put something next to something.'

**Diving-48** (Li et al., 2018), which contains ∼18k videos with 48 different diving action classes, is an action recognition dataset that minimizes contextual biases, *i.e.*, scenes or objects.

**FineGym** (Shao et al., 2020) is a fine-grained action dataset built on top of gymnastic videos. We adopt the *Gym288* and *Gym99* sets for experiments that contains 288 and 99 classes, respectively.

**Action recognition architecture.** We employ TSN ResNets (Wang et al., 2016) as 2D CNN backbones and TSM ResNets (Lin et al., 2019) as 3D CNN backbones. TSM enables to obtain the effect of spatio-temporal convolutions using spatial convolutions by shifting a part of input channels along the temporal axis before the convolution operation. TSM is added into each residual block of the ResNet. We adopt ImageNet pre-trained weights for our backbones. To transform the backbones to the self-similarity network (SELFYNet), we insert a single SELFY block after the third stage in the backbones. For SELFY block, we use the convolution method as a default feature extraction method and use multi-channel $1 \times 3 \times 3$ convolution kernels. For more details, please refer Appendix A, B.

**Training & testing.** For training, we sample a clip of 8 or 16 frames from each video by using segment-based sampling (Wang et al., 2016). The spatio-temporal matching region $(L, U, V)$ of SELFY block is set as $(5, 9, 9)$ or $(9, 9, 9)$ when using 8 or 16 frames, respectively. For testing, we sample one or two clips from a video, crop their center, and evaluate the averaged prediction of the sampled clips. For more details, please refer Appendix A.

Table 3: **Ablations on SS-V1**. Top-1 & 5 accuracy (%) are shown.

(a) **Types of similarity**. Performance comparison with different sets of temporal offset in SELFY block. $\{\cdot\}$ denotes a set of temporal offset $l$.

| model | range of $l$ | FLOPs | top-1 | top-5 | model | range of $l$ | FLOPs | top-1 | top-5 |
|-------|------|-------|-------|-------|-------|------|-------|-------|-------|
| TSN-R18 | - | 14.6 | 16.2 | 40.8 | TSM-R18 | - | 14.6 | 43.0 | 72.3 |
| | $\{0\}$ | 15.3 | 16.8 | 42.2 | | $\{0\}$ | 15.3 | 45.0 | 73.4 |
| | $\{1\}$ | 15.3 | 39.7 | 68.9 | | $\{1\}$ | 15.3 | 47.1 | 76.3 |
| SELFYNet | $\{-1, 0, 1\}$ | 16.3 | 44.7 | 73.9 | SELFYNet | $\{-1, 0, 1\}$ | 16.3 | 47.8 | 76.7 |
| | $\{-2, \cdots, 2\}$ | 17.3 | **46.9** | 75.9 | | $\{-2, \cdots, 2\}$ | 17.3 | 48.4 | 77.6 |
| | $\{-3, \cdots, 3\}$ | 18.3 | **46.9** | **76.2** | | $\{-3, \cdots, 3\}$ | 18.3 | **48.6** | **77.7** |

(b) **Feature extraction and integration methods.** Smax denotes the soft-argmax operation. MLP consist of four fully connected (FC) layers. The $1 \times 1 \times 1$ convolutional layer in the feature integration stage is omitted from this table.

| model | feature extraction | feature integration | top-1 | top-5 |
|-------|--------------------|--------------------|-------|-------|
| TSM-R18 | - | - | 43.0 | 72.3 |
| | Smax | FC | 44.0 | 72.3 |
| | MLP | FC | 45.9 | 75.1 |
| SELFYNet | Conv | FC | 46.7 | 75.8 |
| | Conv | MLP | 47.2 | 75.9 |
| | Conv | Conv | **48.4** | **77.6** |

## 4.2 COMPARISON WITH THE STATE-OF-THE-ART METHODS

For a fair comparison, we compare our model with other models that are not pre-trained on additional large-scale video datasets, *e.g.*, Kinetics (Kay et al., 2017) or Sports1M (Karpathy et al., 2014) in the following experiments.

Table 1 summarizes the results on SS-V1&V2. The first and second compartment of the table shows the results of other 2D CNN and (pseudo-) 3D CNN models, respectively. The last part of each compartment shows the results of SELFYNet. SELFYNet with TSN-ResNet (SELFYNet-TSN-R50) achieves 50.7% and 62.7% at top-1 accuracy, respectively, which outperforms other 2D models using 8 frames only. When we adopt TSM ResNet (TSM-R50) as our backbone and use 16 frames, our method (SELFYNet-TSM-R50) achieves 54.3% and 65.7% at top-1 accuracy, respectively, which is the best among the single models. Compared to TSM-R50, a single SELFY block obtains a significant gain of 7.0%p and 4.5%p at top-1 accuracy, respectively; our method is more accurate than TSM-R50 Two-stream on both datasets. Finally, our ensemble model (SELFYNet-TSM-R50$_{EN}$) with 2-clip evaluation sets a new state of the art on both datasets by achieving 56.6% and 67.7% at top-1 accuracy, respectively.

Table 2 summarizes the results on Diving-48 & FineGym. For Diving-48, TSM-R50 using 16 frames shows 38.8% in top-1 accuracy in our implementation. SELFYNet-TSM-R50 outperforms TSM-R50 by 2.8%p in accuracy so that it sets a new state-of-the-art top-1 accuracy as 41.6% on Diving-48. For FineGym, SELFYNet-TSM-R50 achieves 49.5% and 87.7% at given 288 and 99 classes, respectively, surpassing all the other models reported in Shao et al. (2020).

## 4.3 ABLATION STUDIES

We conduct ablation experiments to demonstrate the effectiveness of the proposed method. All experiments are performed on SS-V1 by using 8 frames. Unless otherwise specified, we set ImageNet pre-trained TSM ResNet-18 (TSM-R18) with the single SELFY block of which $(L, U, V) = (5, 9, 9)$, as our default SELFYNet.

**Types of similarity.** In Table 3a, we investigate the effect of different types of similarity by varying the set of temporal offset $l$ on both TSN-ResNet-18 (TSN-R18) and TSM-R18. Interestingly, learning spatial self-similarity ($\{0\}$) improves accuracy on both backbones, which implies that self-similarity features help capture structural patterns of visual features. Learning cross-similarity with a short temporal range ($\{1\}$) shows a noticeable gain in accuracy on both backbones, indicating the significance of motion features. Learning STSS outperforms other types of similarity, and the accuracy of SELFYNet increases as the temporal range becomes longer. When STSS takes a far-sighted

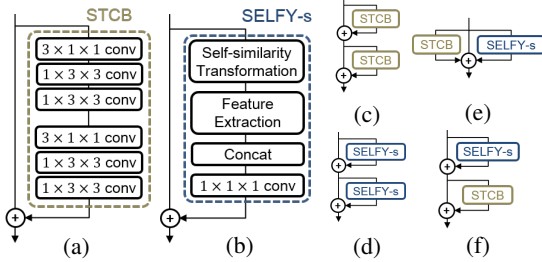

Figure 4: **Basic blocks and their combinations**. (a) spatio-temporal convolution block (STCB), (b) SELFY-s block, and (c-f) their combinations.

Table 5: **Spatio-temporal features v.s. STSS features**. Results of different combinations of two blocks ((a) - (f) from Fig. 4) are shown.

| model, TSN-R18 | top-1 | top-5 |
|---|---|---|
| baseline | 16.2 | 40.8 |
| (a) STCB | 42.4 | 71.7 |
| (b) SELFY-s | 46.3 | 75.1 |
| (c) STCB + STCB | 44.4 | 73.7 |
| (d) SELFY-s + SELFY-s | 46.8 | 75.9 |
| (e) SELFY-s + STCB (parallel) | 46.9 | 76.5 |
| (f) SELFY-s + STCB (sequential) | **47.6** | **76.6** |

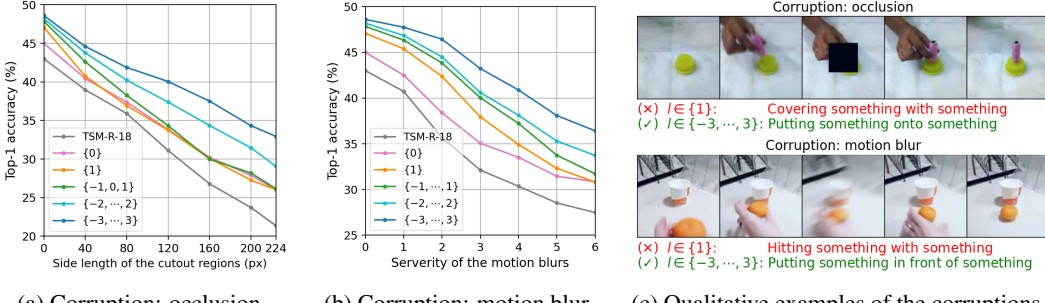

(a) Corruption: occlusion. (b) Corruption: motion blur. (c) Qualitative examples of the corruptions.

Figure 5: **Results of the robustness experiment**. (a) and (b) show top-1 accuracy of SELFYNet variants (Table 3a) with different occlusions and motion blurs, respectively. (c) shows qualitative examples that SELFYNet ($\{-3, \cdots, 3\}$) answers correct, while SELFYNet ($\{1\}$) fails.

view on motion, STSS learns both short-term and long-term interactions in videos, as well as spatial self-similarity. Qualitative results in Appendix D show that SELFYNet with a long temporal range ($\{-3, \cdots, 3\}$) effectively captures long-term interactions in videos.

**Feature extraction and integration methods.** In Table 3b, we compare the performance of different combinations of feature extraction and integration methods. From the 2nd to the 4th rows, different feature extraction methods are compared fixing the feature integration methods to a single fully-connected (FC) layer. Compared to the baseline, the use of soft-argmax, which extracts spatial displacement features, improves top-1 accuracy by 1.0%p. Replacing soft-argmax with MLP provides an additional gain of 1.9%p at top-1 accuracy, showing the effectiveness of directly using similarity values. When using the convolution method for feature extraction, we achieve 46.7% at top-1 accuracy; the multi-channel convolution kernel is more effective in capturing structural patterns along $(u, v)$ dimensions than MLP. From the 4th to the 6th rows, different feature integration methods are compared fixing the feature extraction method to convolution. Replacing the single FC layer with MLP improves the top-1 accuracy by 0.6%p. Replacing MLP with convolutional layers further improves and achieves 48.4% at top-1 accuracy. These results demonstrate that our design choice of using convolutions along $(u, v)$ and $(h, w)$ dimensions is the most effective in learning the geometry-aware STSS representation. For more ablation experiments, please refer to Appendix B.

## 4.4 COMPLEMENTARITY BETWEEN SPATIO-TEMPORAL FEATURES AND STSS FEATURES

We conduct experiments for analyzing different meanings of spatio-temporal features and STSS features. We organize two basic blocks for representing two different features: spatio-temporal convolution block (STCB) that consists of several spatial-temporal convolutions (Fig. 4a) and SELFY-s block, light-weighted version of the SELFY block by removing spatial convolution layers (Fig. 4b). Both blocks have the same receptive fields and a similar number of parameters for a fair comparison. Different combinations of the basic blocks are inserted after the third stage of TSN-ResNet-18. Table 5 summarizes the results on SS-V1. STSS features (Fig. 4b, 4d) are more effective than spatio-temporal features (Fig. 4a, 4c) in top-1 and top-5 accuracy when the same number of blocks are inserted. Interestingly, the combination of two different features (Fig. 4e, 4f) shows better results in top-1 and top-5 accuracy compared to the single feature cases (Fig. 4c, 4d), which demonstrate

that both features complement each other. We conjecture that this complementarity comes from different characteristics of the two features; while spatio-temporal features are obtained by directly encoding appearance features, STSS features are obtained by suppressing variations in appearance and focusing on the relational features in space and time.

## 4.5 IMPROVING ROBUSTNESS WITH STSS

In this section, we demonstrate that STSS representation helps video-processing models to be more robust to video corruptions. We test two corruptions that are likely to happen in the real world videos: occlusion and motion blur. To induce the corruptions, We either cut out a rectangle patch of a particular frame or generate a motion blur (Hendrycks & Dietterich, 2019). We corrupt a single center-frame for every clip of SS-V1 at the testing phase and gradually increase the severity of corruptions. We compare the results of TSM-R18 and SELFYNet variants of Table 3a. Fig. 5a, 5b summarizes the results of two corruptions, respectively. The top-1 accuracy of TSM-R18 and SELF-YNets with the short temporal range ($\{0\}$, $\{1\}$, and $\{-1, 0, 1\}$) significantly drops as the severity of corruptions becomes harder. We conjecture that the features of the corrupted frame propagate through the stacked TSMs, confusing the entire network. However, the SELFYNets with the long temporal range ($\{-2, \cdots, 2\}$ and $\{-3, \cdots, 3\}$) show more robust performance than the other models. As shown in Fig. 5a, 5b, the accuracy gap between SELFYNets with the long temporal range and the others increases as the severity of corruptions becomes higher, indicating that the larger size of STSS features can improve the robustness on action recognition. We also present some qualitative results (Fig. 5c) where two SELFYNets with different temporal ranges, $\{1\}$ and $\{-3, \cdots, 3\}$, both answer correctly without corruption, while the SELFYNet with $\{1\}$ fails for the corrupted input.

## 5 CONCLUSION

In this paper, we have proposed to learn a generalized, far-sighted motion representation from STSS for video understanding. The comprehensive analyses on the STSS demonstrate that STSS features effectively capture both short-term and long-term interactions, complement spatio-temporal features, and improve the robustness of video-processing models. Our method outperforms other state-of-the-art methods on the three benchmarks for video action recognition.

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

Table 6: **TSN-ResNet & TSM-ResNet backbone**.

| Layers | TSN-ResNet-18 | TSM-ResNet-18 | TSN-ResNet-50 | TSM-ResNet-50 | Output size |
|---|---|---|---|---|---|
| $conv_1$ | $1\times7\times7$, 64, stride 1,2,2 | | | | $T\times112\times112$ |
| $pool_1$ | $1\times3\times3$ max pool, stride 1,2,2 | | | | $T\times56\times56$ |
| $res_2$ | $\begin{bmatrix}1\times3\times3,\ 64\\1\times3\times3,\ 64\end{bmatrix}\times2$ | $\begin{bmatrix}\text{TSM}\\1\times3\times3,\ 64\\1\times3\times3,\ 64\end{bmatrix}\times2$ | $\begin{bmatrix}1\times1\times1,\ 256\\1\times3\times3,\ 256\\1\times1\times1,\ 256\end{bmatrix}\times3$ | $\begin{bmatrix}\text{TSM}\\1\times1\times1,\ 256\\1\times3\times3,\ 256\\1\times1\times1,\ 256\end{bmatrix}\times3$ | $T\times56\times56$ |
| $res_3$ | $\begin{bmatrix}1\times3\times3,\ 128\\1\times3\times3,\ 128\end{bmatrix}\times2$ | $\begin{bmatrix}\text{TSM}\\1\times3\times3,\ 128\\1\times3\times3,\ 128\end{bmatrix}\times2$ | $\begin{bmatrix}1\times1\times1,\ 512\\1\times3\times3,\ 512\\1\times1\times1,\ 512\end{bmatrix}\times4$ | $\begin{bmatrix}\text{TSM}\\1\times1\times1,\ 512\\1\times3\times3,\ 512\\1\times1\times1,\ 512\end{bmatrix}\times4$ | $T\times28\times28$ |
| $res_4$ | $\begin{bmatrix}1\times3\times3,\ 256\\1\times3\times3,\ 256\end{bmatrix}\times2$ | $\begin{bmatrix}\text{TSM}\\1\times3\times3,\ 256\\1\times3\times3,\ 256\end{bmatrix}\times2$ | $\begin{bmatrix}1\times1\times1,\ 1024\\1\times3\times3,\ 1024\\1\times1\times1,\ 1024\end{bmatrix}\times6$ | $\begin{bmatrix}\text{TSM}\\1\times1\times1,\ 1024\\1\times3\times3,\ 1024\\1\times1\times1,\ 1024\end{bmatrix}\times6$ | $T\times14\times14$ |
| $res_5$ | $\begin{bmatrix}1\times3\times3,\ 512\\1\times3\times3,\ 512\end{bmatrix}\times2$ | $\begin{bmatrix}\text{TSM}\\1\times3\times3,\ 512\\1\times3\times3,\ 512\end{bmatrix}\times2$ | $\begin{bmatrix}1\times1\times1,\ 2048\\1\times3\times3,\ 2048\\1\times1\times1,\ 2048\end{bmatrix}\times3$ | $\begin{bmatrix}\text{TSM}\\1\times1\times1,\ 2048\\1\times3\times3,\ 2048\\1\times1\times1,\ 2048\end{bmatrix}\times3$ | $T\times7\times7$ |
| | global average pool, FC | | | | # of classes |

## A  IMPLEMENTATION DETAILS

**Architecture details.** We use TSN-ResNet and TSM-ResNet as our backbone (see Table 6) and initialize them with ImageNet pre-trained weights. We insert a single SELFY block into right after $res_3$ and use the convolution method as a default feature extraction method. We set the spatio-temporal matching region of SELFY block, $(L, U, V)$, as $(5, 9, 9)$ or $(9, 9, 9)$ when using 8 or 16 input frames, respectively. We stack four $1 \times 3 \times 3$ convolution layers along $(l, u, v)$ dimension for the feature extraction method, and use four $3 \times 3$ convolution layer along $(x, y)$ dimension for the feature enhancement. We reduce a spatial resolution of video feature tensor, $\mathbf{V}$, as $14\times14$ for computation efficiency before the self-similarity transformation. After the feature enhancement, we upsample the enhanced feature tensor, $\mathbf{G}^\star$, as $28\times28$ for the residual connection.

**Training.** We sample a clip of 8 or 16 frames from each video by using segment-based sampling (Wang et al., 2016). We resize the sampled clips into $240 \times 320$ images and apply random scaling and horizontal flipping for data augmentation. When applying the horizontal flipping on SS-V1&V2 (Goyal et al., 2017b), we do not flip clips of which class labels include 'left' or 'right' words; the action labels, *e.g.*, 'pushing something from left to right.' We fit the augmented clips into a spatial resolution of $224 \times 224$. For SS-V1&V2, we set the initial learning rate to 0.01 and the training epochs to 50; the learning rate is decayed by 1/10 after $30^{th}$ and $40^{th}$ epochs. For Diving-48 (Li et al., 2018) and FineGym (Shao et al., 2020), we use a cosine learning rate schedule (Loshchilov & Hutter, 2016) with the first 10 epochs for gradual warm-up (Goyal et al., 2017a). We set the initial learning rate to 0.01 and the training epochs to 30 and 40, respectively.

**Testing.** Given a video, we sample 1 or 2 clips, resize them into $240 \times 320$ images, and crop their centers as $224 \times 224$. We evaluate an average prediction of the sampled clips. We report top-1 and top-5 accuracy for SS-V1&V2 and Diving-48, and mean-class accuracy for FineGym.

**Frame corruption details.** We adopt two corruptions, occlusion and motion blur, to test the robustness of SELFYNet. We only corrupt a single center-frame for every validation clip of SS-V1; we corrupt the $4^{th}$ frame amongst 8 input frames. For the occlusion, we cut out a rectangle region from the center of the frame. For the motion blur, we adopt ImageNet-C implementation, which is available online[1]. We set 6 levels of severity for each corruption. We set the side length of the occluded region as 40px, 80px, 120px, 160px, 200px and 224px from the level 1 to 6. For the motion blur, we set (*radius*, *sigma*) tuple arguments as $(15, 5)$, $(10, 8)$, $(15, 12)$, $(20, 15)$, $(25, 20)$, and $(30, 25)$, respectively.

Table 7: **Additional ablations on SS-V1**. Top-1 & 5 accuracy (%) are shown.

(a) **Performance comparison with non-local methods.** NL and CP denote a non-local block and a CP module, respectively.

| model | FLOPs | top-1 | top-5 |
|---|---|---|---|
| TSM-R18 | 14.6G | 43.0 | 72.3 |
| TSM-R18 + NL | 24.8G | 43.8 | 73.1 |
| TSM-R18 + CP | 25.6G | 46.7 | 75.7 |
| SELFYNet | 17.3G | **48.4** | **77.6** |

(b) **Performance comparison with MSNet.** KS and CM denote the kernel soft-argmax and confidence map, respectively.

| model | extraction | $(L, U, V)$ | top-1 | top-5 |
|---|---|---|---|---|
| TSM-R18 | - | - | 43.0 | 72.3 |
| SELFYNet | KS + CM | $(1, 9, 9)$ | 46.1 | 75.3 |
| | KS + CM | $(5, 9, 9)$ | 47.4 | 76.8 |
| | Conv | $(1, 9, 9)$ | 47.1 | 76.3 |
| | Conv | $(5, 9, 9)$ | **48.4** | **77.6** |

(c) **Multi-channel** $3 \times 3 \times 3$ **kernel for feature extraction.** Four convolution layers are used for extracting STSS features. $\{\cdot\}$ denotes a set of temporal offsets $l$.

| model | range of $l$ | top-1 | top-5 |
|---|---|---|---|
| TSM-R18 | - | 43.0 | 72.3 |
| SELFYNet | $\{-1, 0, 1\}$ | 47.4 | 77.0 |
| | $\{-2, \cdots, 2\}$ | 48.3 | 77.2 |
| | $\{-3, \cdots, 3\}$ | **48.5** | **77.4** |

(d) **Spatial matching region**. Performance comparison with different spatial matching-regions, $(U \times V)$.

| model | $U \times V$ | FLOPs | top-1 | top-5 |
|---|---|---|---|---|
| TSM-R18 | - | 14.6 | 43.0 | 72.3 |
| SELFYNet | $5 \times 5$ | 17.1 | 47.8 | 77.1 |
| | $9 \times 9$ | 17.3 | 48.2 | 77.5 |
| | $13 \times 13$ | 18.4 | 48.4 | 77.8 |
| | $17 \times 17$ | 19.8 | **48.6** | **78.3** |

(e) **STSS with visual features V. R** denotes the STSS features, *i.e.*, $\mathrm{ReLU}(\mathbf{F}^{\star} \times_5 \mathbf{W}_{\theta})$ in Eq. 9.

| model | $\mathbf{Z}$ | top-1 | top-5 |
|---|---|---|---|
| TSM-R18 | $\mathbf{V}$ | 43.0 | 72.3 |
| SELFYNet | $\mathbf{R}$ | 45.5 | 75.9 |
| | $\mathbf{R} + \mathbf{V}$ | **48.4** | **77.6** |

(f) **Position**. Performance comparison with different positions of SELFY block. For the last row, 3 SELFY blocks are used in total.

| model | position | top-1 | top-5 |
|---|---|---|---|
| TSM-R18 | - | 43.0 | 72.3 |
| SELFYNet | $\mathrm{pool}_1$ | 45.7 | 77.6 |
| | $\mathrm{res}_2$ | 47.2 | 76.6 |
| | $\mathrm{res}_3$ | 48.2 | 77.5 |
| | $\mathrm{res}_4$ | 46.6 | 76.0 |
| | $\mathrm{res}_5$ | 42.8 | 72.6 |
| | $\mathrm{res}_{2,3,4}$ | **48.6** | **77.9** |

# B    ADDITIONAL EXPERIMENTS

We conduct additional ablation experiments to identify the behaviors of the proposed method. All experiments are performed on SS-V1 by using 8 frames. Unless otherwise specified, we set ImageNet pre-trained TSM ResNet-18 (TSM-R18) with a single SELFY block of which $(L, U, V) = (5, 9, 9)$, as our default SELFYNet.

**Comparison with non-local methods.** We compare our method with popular non-local methods (Wang et al., 2018; Liu et al., 2019), which capture the long-range dynamics of videos. While computing the self-similarity values as ours, both methods use them as attention weights for feature aggregation by multiplying them to the visual features (Wang et al., 2018) or aligning top-$K$ corresponding features (Liu et al., 2019); they both do not use STSS itself as a relational representation. In contrast, our method does it indeed and learns a more powerful relational feature from STSS. The difference between our method and non-local methods are illustrated in Fig. 6.

We have conducted experiments for performance comparison, and the results are shown in Table 7a. We re-implement the non-local block and the CP module in Pytorch based on their official codes[23]. For a fair comparison, we insert a single block or module at the same position (after $res_3$ of ResNet-18). Note that our method downsamples a spatial resolution of $\mathbf{V}$ to $14 \times 14$ before STSS transformation, whereas the others do not. Compared to the non-local block and the CP module, SELFY block improves top-1 accuracy by 4.4%p and 1.5%p, while computing less floating-point operations as 7.5 GFLOPs and 8.3 GFLOPs, respectively. It demonstrates that the direct integration of

---

[1]https://github.com/hendrycks/robustness
[2]https://github.com/xingyul/cpnet
[3]https://github.com/facebookresearch/video-nonlocal-net

STSS features is more effective for action recognition than the indirect ways of using STSS, *e.g.*, re-weighting visual-semantic features or learning correspondences.

**Comparison with correlation-based methods.** We also compare our method with correlation-based methods (Kwon et al., 2020; Wang et al., 2020). While correlation-based methods extract motion features between two adjacent frames only and are thus limited to short-term motion, our method effectively captures bi-directional and long-term motion information via learning with the sufficient volume of STSS. Our method can also exploit richer information from the self-similarity values than other methods. MS module (Kwon et al., 2020) only focuses on the maximal similarity value of the $(u, v)$ dimensions to extract flow information, and Correlation block (Wang et al., 2020) uses an $1 \times 1$ convolution layer for extracting motion features from the similarity values. In contrast to the two methods, we introduce a generalized motion learning framework using the self-similarity tensor at Sec 3.2 of our main paper. The difference between our method and correlation-based methods are illustrated in Fig. 6.

We also have conducted experiments to compare our method with MSNet (Kwon et al., 2020), one of the correlation-based methods. For an apple-to-apple comparison, we apply kernel soft-argmax and max pooling operation (*KS + CM* in Kwon et al. (2020)) to our feature extraction method by following their official codes[4]. Please note that, when we restrict the temporal offset $l$ to $\{1\}$, the SELFY block using KS + CM is equivalent to the MS module of which *feature transformation* layers are the standard 2D convolutional layers. Table 7b summarizes the results. KS+CM method achieves 46.1% at top-1 accuracy. As we enlarge the temporal window $L$ to 5, we obtain an additional gain as 1.3%p. The learnable convolution layers improve top-1 accuracy by 1.0%p in both cases. The results demonstrates the effectiveness of learning geometric patterns within the sufficient volume of STSS tensors for learning abundant motion features.

**Multi-channel $3 \times 3 \times 3$ kernel for feature extraction.** We investigate the effect of the convolution method for STSS feature extraction, when we use multi-channel $3 \times 3 \times 3$ kernels. For the experiment, we stack four $3 \times 3 \times 3$ convolution layers followed by the feature integration step, which are the same as in Section 3.2.2. Table 7c summarizes the results. Note that we do not report models of which temporal window $L = 1$, *e.g.*, $\{0\}$ and $\{1\}$. As shown in the table, indeed, the long temporal range gives the higher accuracy. However, the effect of the $3 \times 3 \times 3$ kernel is comparable to that of the $1 \times 3 \times 3$ kernel in Table 3a in terms of accuracy. Considering the accuracy-computation trade-off, we choose to fix the kernel size, $L_\kappa \times U_\kappa \times V_\kappa$, as $1 \times 3 \times 3$ for the STSS feature extraction.

**Spatial matching region.** In Table 7d, we compare a single SELFY block with different spatial matching regions, $(U, V)$. As a result, indeed, the larger spatial matching region leads the better accuracy. Considering the accuracy-computation trade-off, we set our spatial matching region, $(U, V)$, as $(9, 9)$ as a default.

**Fusing STSS with visual features.** We evaluate SELFYNet purely based on STSS features to see how much the ordinary visual feature $\mathbf{V}$ contributes for the final prediction. That is, we pass the STSS features, $\mathrm{ReLU}(\mathbf{F}^\star \times_5 \mathbf{W}_\theta)$, into the downstream layers without the visual features $\mathbf{V}$ (Eq. 9 in our main paper). For the simplicity of description, we denote the relational feature $\mathrm{ReLU}(\mathbf{F}^\star \times_5 \mathbf{W}_\theta)$ by $\mathbf{R}$ . Table 7e compares the results of using different cases of the output tensor $\mathbf{Z}$ ($\mathbf{Z} = \mathbf{V}$, $\mathbf{Z} = \mathbf{R}$, and $\mathbf{Z} = \mathbf{R} + \mathbf{V}$) on SS-V1. Interestingly, SELFYNet using only $\mathbf{R}$ achieves 45.5% at top-1 accuracy, which is higher as 2.5%p than the baseline. As we add $\mathbf{V}$ to $\mathbf{R}$, we obtain an additional gain of 2.9%p. It indicates that the STSS features and the visual features are complementary to each other.

**Block position.** From the 2nd to the 6th row of Table 7f, we identify the effect of different positions of SELFY block in the backbone. We resize the spatial resolution of the video tensor, $(X, Y)$, into $14 \times 14$, and fix the matching region, $(L, U, V)$, as $(5, 9, 9)$ for all the cases maintaining the similar computational cost. SELFY after the $res_3$ shows the best trade-off by achieving the highest accuracy among the cases. The last row in Table 7f shows that the multiple SELFY blocks improve accuracy compared to the single block.

---

[4]https://github.com/arunos728/MotionSqueeze

Table 8: **Performance comparison with the local self attention mechanisms**. R.P.E. is an abbreviation for relative positional embeddings. Top-1, 5 accuracy (%) are shown.

| model | similarity | extraction | top-1 | top-5 |
|---|---|---|---|---|
| TSM-R18 | - | - | 43.0 | 72.3 |
| SELFYNet | embedded Gaussian | multiplication w/ $\mathbf{V}$ + R.P.E. | 43.8 | 72.3 |
| | embedded Gaussian | Conv | 47.6 | 76.8 |
| | cosine | Conv | 47.8 | 77.1 |

## C  THE RELATIONSHIP WITH THE LOCAL SELF-ATTENTION MECHANISMS

The local self-attention (Hu et al., 2019; Ramachandran et al., 2019; Zhao et al., 2020) and our method have a common denominator of using the self-similarity tensor but use it in a very different way and purpose. The local self-attention mechanism aims to aggregate the local context features using the self-similarity tensor and it thus uses the self-similarity values as attention weights for feature aggregation. However, our method aims to learn a generalized motion representation from the local STSS, so the final STSS representation is directly fed into the neural network instead of multiplying it to local context features.

For an empirical comparison, we conduct an ablation experiment as follows. We extend the local self attention layer (Ramachandran et al., 2019) to temporal dimension, and then add the *spatio-temporal* local self-attention layer, which is followed by feature integration layers, after $res_3$. All experimental details are the same as those in Appendix A, except that we reduce the channel dimension $C'$ of appearance feature $\mathbf{V}$ to 32. Table 8 summarizes the results on SS-V1. The spatio-temporal local self-attention layer is accurate as 43.8% at top-1 accuracy, and both of SELFY blocks using the embedded Gaussian and the cosine similarity outperform the local self-attention by achieving top-1 accuracy as 47.6% and 47.8%, respectively. These results are in alignment with the prior work (Liu et al., 2019), which reveals that the self-attention mechanism hardly captures motion features in video.

## D  VISUALIZATIONS

In Fig. 7, we visualize some qualitative results of two different SELFYNet-TSM-R18 ($\{1\}$ and $\{-3, \cdots, 3\}$) on SS-V1. We show the different predictions of the two models with 8 input frames. We also overlay Grad-CAMs (Selvaraju et al., 2017) on the input frames to see whether a larger volume of STSS benefits to capture long-term interactions in videos. We take Grad-CAMs of features which is right before a global average pooling layer. As shown in the figure, the STSS with the sufficient volume helps to learn more enriched context of temporal dynamics in the video; in Fig. 7a, for example, SELFYNet with the range of ($\{-3, \cdots, 3\}$) focuses on not only regions on which an action occurs but also focuses on the white-stain after the action to verify whether the stain is wiped off or not.

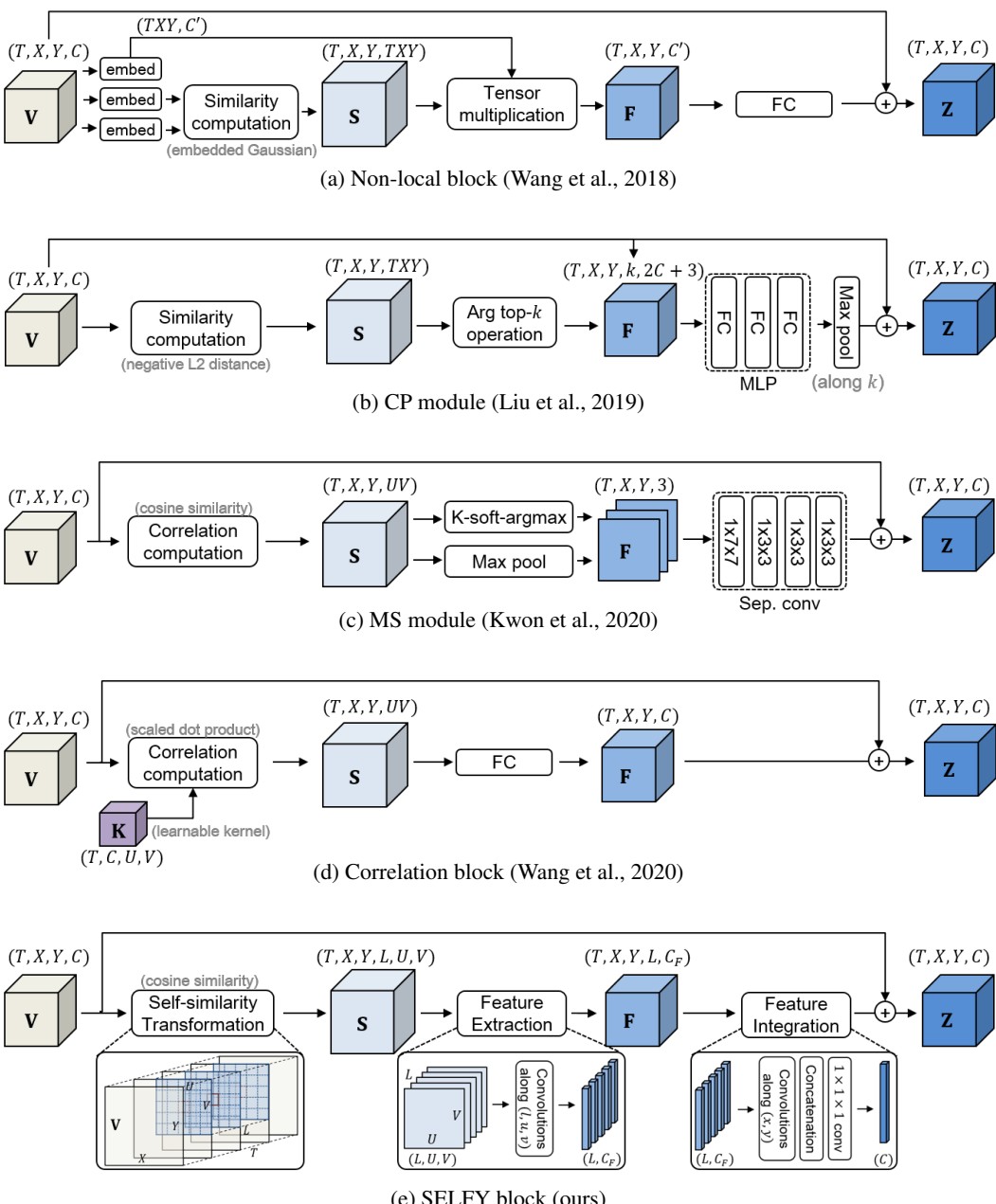

Figure 6: **Comparison with non-local approaches (Wang et al., 2018; Liu et al., 2019) and correlation-based methods (Kwon et al., 2020; Wang et al., 2020).** From the top, non-local block, CP module, MS module, Correlation block and SELFY block are illustrated.

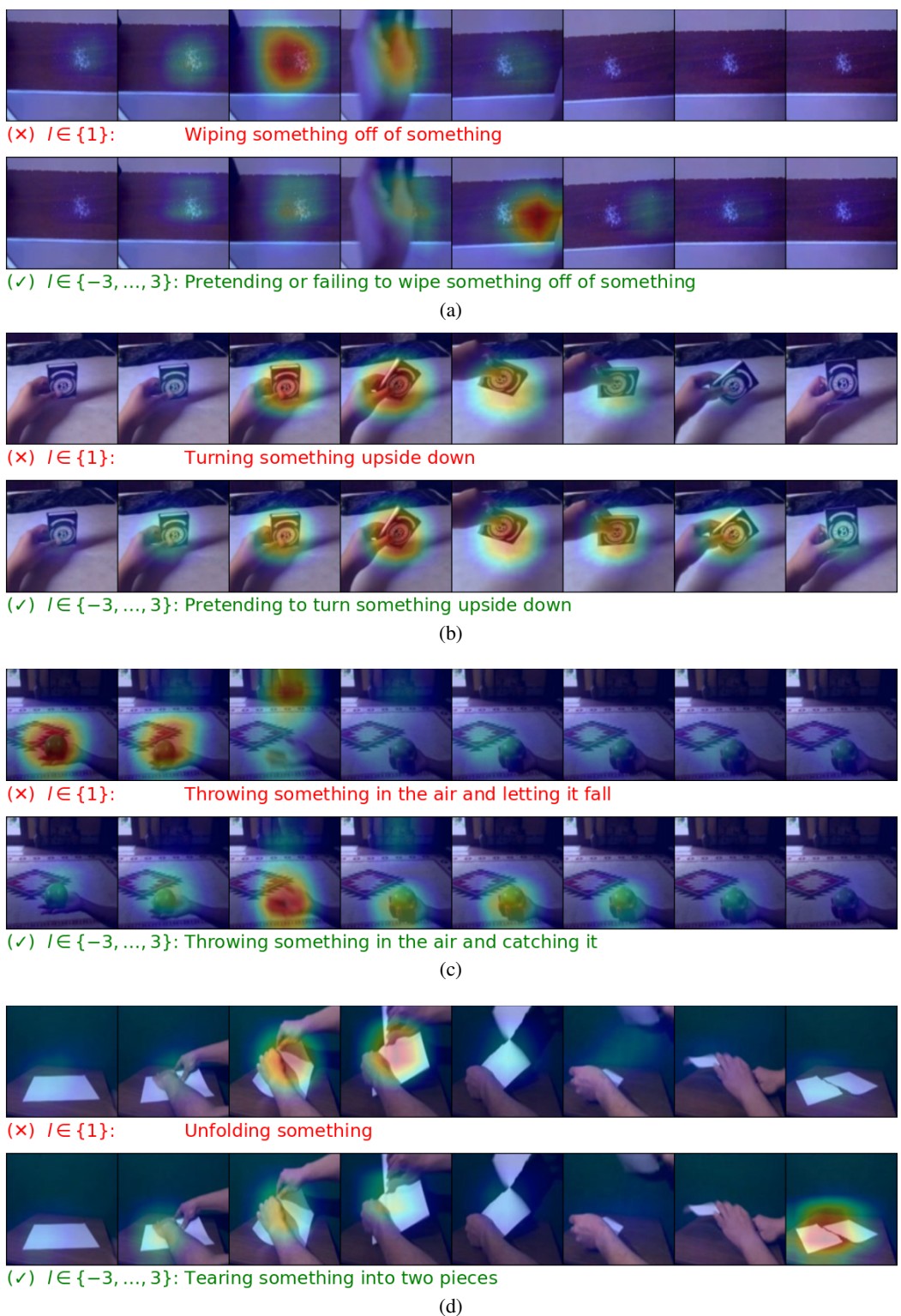

Figure 7: **Qualitative results** of two SELFYNets on SS-V1. Each subfigure visualizes prediction results of the two models with Grad-CAM-overlaid RGB frames. The correct and wrong predictions are colorized as green and red, respectively.

