# OpenReview forum: "Learning Self-Similarity in Space and Time as a Generalized Motion for Action Recognition"
_ICLR.cc/2021/Conference — Reject_

### Official Review · AnonReviewer1 · 2020-10-15
**Nice experiments with limited novelty**

**Rating:** 5
**Confidence:** 3

**Review:**

The paper proposes a novel nerual block based on the classifical spatio-temporal self-similarity (STSS), named SELFY, which can be easily inserted into nerual architectures and learned end-to-end without additional supervision. SELFY could capture long-term interaction and fast motion in the video with a sufficient volume  of the neighborhood in time and space.
The nice point of the method is that it is heavily investigtated through experiments. Evaluated on three standard action recognition benchmark datasets,  the proposed SELFY is demonstrated the superiority over previous methods for motion modeling as well as the complementarity to spatio-temporal features from direct convolution. Moreover, the paper is clear and seems correct ,technically.

comments:

-The first concern is about  the limited novelty of the method. Despite revisting the self-similarity, from section 3.1, the learning of generalized and long-term information is a property of the STSS rather than a contribution of this work. If necessary, authors should give more details on the classifical STSS to show the improvement.
-The authors briefly mention the differences between the new proposed and non-local approaches (Wang et al., 2018; Liu et al., 2019) and correlation-based methods (Wang et al., 2020; Kwon et al., 2020), which appears inadequate and is better to integrate them into a comparison in the form of Figure 1.
-How are parameters (5, 9, 9) and (9, 9 ,9) chosen? The best result occurs as the  temporal offsets being chosen from {-3, ..., 3}. Will there be better accuracy when the range is larger?

---

> ### Author Response · Authors · 2020-11-19
> **Response to Reviewer 1 (2/2)**
>
> > [R1] The authors briefly mention the differences between the new proposed and non-local approaches (Wang et al., 2018; Liu et al., 2019) and correlation-based methods (Wang et al., 2020; Kwon et al., 2020), which appears inadequate and is better to integrate them into a comparison in the form of Figure 1.
>
> While computing the self-similarity values as ours, the non-local approaches [3,4] use them as attention weights for feature aggregation by multiplying them to the visual features [3] or aligning top-$K$ corresponding features [4]; they both do not use STSS itself as a relational representation. In contrast, our method does it indeed and learns a more powerful relational feature from STSS. As for the performance comparison, Table 7a of our appendix shows that our method clearly outperforms non-local approaches in top-1, top-5 accuracy and FLOPs.
>
> Furthermore, while the correlation-based methods [5,6] extract motion features between two adjacent frames only and are thus limited to short-term motion, our method effectively captures bi-directional and long-term motion information via learning with the sufficient volume of STSS. Our method can also exploit richer information from the self-similarity values than other methods. MS module [5] only focuses on the maximal similarity value of the $(U,V)$ dimension to extract flow information, and Correlation block [6] uses an $1 \times 1$  convolution layer for extracting motion features from the similarity values. In contrast to the two methods, we introduce a generalized motion learning framework using the self-similarity tensor at Sec 3.2 of our paper and demonstrate the effectiveness of our method through ablation studies.
>
> Following the reviewers’ suggestion, we will add a figure that describes these differences between the previous ones and ours in our final manuscript.
>
> [3] X. Wang, R. Girshick, A. Gupta, and K. He. “Non-local neural networks.” CVPR. 2018.\
> [4] X. Liu, J.Y. Lee, and H. Jin. “Learning video representations from correspondence proposals.” CVPR. 2019.\
> [5] H. Wang, D. Tran, L. Torrensani, and M. Feiszli. “Video modeling with correlation networks.” CVPR. 2020.\
> [6] H. Kwon, M. Kim, S. Kwak, and M. Cho. “MotionSqueeze: neural motion feature learning for video understanding.” ECCV. 2020.
>
> > [R1] How are parameters (5, 9, 9) and (9, 9 ,9) chosen? The best result occurs as the temporal offsets being chosen from {-3, ..., 3}. Will there be better accuracy when the range is larger?
>
> We set the matching region $(L,U,V)$ to $(5,9,9)$ and $(9,9,9)$ since they perform the best accuracy-computation trade-offs. When we use 8 frames as the input, the accuracy saturates as the $(L,U,V)$ volume becomes larger than $(5,9,9)$, resulting in worse trade-offs. When increasing the input frames from 8 to 16, we simply extend $L$ to 9 maintaining the time span of STSS. Table B summarizes the SS-V1 results of FLOPs and accuracies varying $(L,U,V)$ given 8 RGB frames.
>
> Table B: **Performance comparison with different matching regions of SELFY block.**
>
> |model|$(L,U,V)$|FLOPs (G)|top-1|top-5|
> |:---|:---:|:---:|:---:|:---:|
> |TSM-R18 |-|14.6|43.0|72.3|
> |SELFYNet|$(1,9,9)$|15.3|47.1|76.3|
> |SELFYNet|$(3,9,9)$|16.3|47.8|76.7|
> |SELFYNet|$(5,9,9)$|17.3|48.4|77.6|
> |SELFYNet|$(7,9,9)$|18.3|**48.6**|77.7|
> |SELFYNet|$(11,9,9)$|20.2|48.4|77.6|
> |SELFYNet|$(5,5,5)$|17.1|47.8|77.1|
> |SELFYNet|$(5,13,13)$|18.4|48.4|77.8|
> |SELFYNet|$(5,17,17)$|19.8|**48.6**|**78.3**|

---

> ### Author Response · Authors · 2020-11-19
> **Response to Reviewer 1 (1/2)**
>
> Thanks for your constructive comments, we will revise our paper by reflecting all of them as much as possible.
>
> > [R1] The first concern is about the limited novelty of the method. Despite revisiting the self-similarity, from section 3.1, the learning of generalized and long-term information is a property of the STSS rather than a contribution of this work. If necessary, authors should give more details on the classifical STSS to show the improvement.
>
>
> While the concept of STSS is already proposed in the prior work [1,2], there are important differences in the way of using STSS. First of all, note that no representation learning is involved in the prior work. E.g., in [1], they extract a pixel-wise hand-crafted STSS feature from the correlation volume $(L,U,V)$ by transforming the correlation volume into a binned log-polar representation and selecting the maximal correlation value in each bin. In contrast, the main contribution of our work is to use STSS as a basis for end-to-end learning a powerful long-term relational representation rather than using STSS feature itself as in the prior work. This is motivated by our novel view on the conventional motion (i.e., optical flow) as one-hot cross-similarity, which is a form of sparse and temporally-restricted STSS as briefly mentioned in our introduction. To this end, we explore different representation learning strategies (Sec.3.2.1 & 3.2.2) and propose our model that consists of convolutions along the $(L,U,V)$ dimension (Sec.3.2.1), which learn relational patterns among similarity values, and spatial convolutions (Sec 3.2.2), which integrate pixel-wise STSS features by extending the receptive fields.
>
> Furthermore, unlike the previous methods, we propose to combine it with the ordinary visual feature (via addition in Eq.(9)) as a complementary one. This is also important. How complementary are they to each other? We find a clear synergic effect of combining STSS with visual features. Denoting the relational feature $\mathrm{ReLU}(\mathbf{F}^\star \times_5 \mathbf{W}_\theta)$ by $\mathbf{R}$ and the ordinary visual feature by $\mathbf{V}$, Table A below compare the cases of $\mathbf{V}$, $\mathbf{R}$, and $\mathbf{R}+\mathbf{V}$ on SS-V1. As we use the combination of $\mathbf{V}$ and $\mathbf{R}$, we obtain the additional gain of 2.9 \%p.
>
> Table A: **The effect of STSS representation ($\mathbf{R}$) with the ordinary visual feature ($\mathbf{V}$)**
>
> |model|feature|top-1|top-5|
> |:---|:---:|:---:|:---:|
> |TSM-R18|$\mathbf{V}$|43.0|72.3|
> |SELFYNet|$\mathbf{R}$|45.5|75.9|
> |SELFYNet (ours)|$\mathbf{R}$ + $\mathbf{V}$|**48.4**|**77.6**|
>
> [1] E. Shechtman  and M. Irani. “Matching local self-similarities across images and videos.” CVPR. 2007.\
> [2] I.N Junejo, E. Dexter, I. Laptev, and P. Perez. “Cross-view action recognition from
> temporal self-similarities.” ECCV. 2008.

---

### Official Review · AnonReviewer3 · 2020-10-22
**This is an interesting work utilized STSS to help learn representation end-to-end and achieved good empirical results on several action recognition benchmarks.**

**Rating:** 6
**Confidence:** 4

**Review:**

### Summary:

This submission proposed a motion representation method based on spatio-temporal self-similarity (STSS), which represents each local region as similarities to its neighbors in both spatial and temporal dimension. There are previous works (e.g., Ref[1] , [2], [5] listed here) which utilize STSS for feature extractions, authors claim that this work is the first one to learn STSS representation based on modern CNN architecture. The proposed method is implemented as a neural block, i.e., SELFY, which can be applied into neural architectures and learned end-to-end without additional supervision. On 3 standard human action recognition data sets, Something-Something-V1 & V2, Diving-48, and FineGym, the proposed method achieves quite good empirical results.

### Strengths & Originality:

The idea of utilizing patio-temporal self-similarity (STSS) for feature representation in the modern CNN framework for human action recognition is interesting. I also like the concept that "fix our attention to the similarity map to the very next frame within STSS and attempt to extract a single displacement vector to the most likely position at the frame, the problem reduces to optical flow ...... In contrast, we leverage the whole volume of STSS ......", as to view the proposed method as a generalized & rich optical flow.

Furthermore, STSS should be helpful for view invariant action recognition, i.e., one of the fundamental challenges from video data recognition.

Empirical results on Something-Something V1 & V2, Diving-48, FineGym show that the proposed achieves the state-of-the-art results, though seems marginally.  For example, the proposed Ensemble model (SELFYNet-TSM-R50EN ) achieved 56.6% and 67.7% attop-1 accuracy for V1 & V2, vs. the MSNet-TSM-R50EN got quite similar performance as 55.1% and 67.1% (Kwon et al., 2020, Ref. [3]).

On FineGym, SELFYNet-TSM-R50 achieves 49.5% and 87.7%, these do look better & more clearly, compared with 46.5%, 81.2% reported from (Shao et al, 2020, Ref. [4]).

### Weakness:

From Figure. 2 or Section 3.2.2, the final STSS representation Z is the combination of original input V and the STSS feature F(S(V)), so kind of unclear how much contribution is made from the original V in terms of the final recognition performance. Maybe this part is addressed in the experimental section somewhere.

The proposed neural block, SELFY, includes 3 parts as self-similarity transformation, feature extraction, and feature integration, it seems none of them is original but the combination of these put into the end-of-end NN structure is new as claimed by authors. There are some additional experiment results presented in section 4.3 (for different types of similarity, and for different feature extraction), still, it seems unclear how important or sensitive for each step within the whole framework based on TSN ResNets and TSM ResNets, given the complexity here.

Self-similarity can be applied for both image and videos (i.e., Section 3 or Ref [5], etc), I know the focus of this submission is video action recognition, still, seems interesting to know the proposed framework "3 steps combined + CNN end-of-end" apply to image object recognition. If that also achieved good results compared with state-of-the-art for object recognition, it will be a strong support for the proposed methodology, and if not, which is also good to know & could be multiple reasons behind.

Besides these video based action benchmarks, it should be interesting to see results on a depth enriched data set (i.e., RGB-D), as missing depth information is one of the limitations from video data. Ideally, we should see similar good performance if the proposed methodology is effective for representation learning.

### Reference:

1. Videos as Space-Time Region Graphs, Xiaolong Wang, Abhinav Gupta, Proc. European Conference on Computer Vision (ECCV) 2018

2. TSM: Temporal Shift Module for Efficient Video Understanding, Ji Lin, Chuang Gan, and Song Han,  Proc. IEEE International Conference on Computer Vision (ICCV), 2019

3. MotionSqueeze: Neural Motion Feature Learning for Video Understanding, Heeseung Kwon, Manjin Kim, Suha Kwak, Minsu Cho, Proc. European Conference on Computer Vision (ECCV) , 2020

4. Dian Shao, Yue Zhao, Bo Dai, and Dahua Lin. Finegym: A hierarchical video dataset for finegrained action understanding. In Proc. IEEE Conference on Computer Vision and Pattern Recognition (CVPR), 2020

5. Imran N. Junejo, Emilie Dexter, Ivan Laptev and Patrick Perez, Cross-View Action Recognition from TemporalSelf-Similarities, Proc. European Conference on Computer Vision (ECCV) 2008

---

> ### Author Response · Authors · 2020-11-19
> **Response to Reviewer 3 (2/2)**
>
> > [R3] Self-similarity can be applied for both image and videos (i.e., Section 3 or Ref [5], etc), I know the focus of this submission is video action recognition, still, seems interesting to know the proposed framework "3 steps combined + CNN end-of-end" apply to image object recognition. If that also achieved good results compared with state-of-the-art for object recognition, it will be a strong support for the proposed methodology, and if not, which is also good to know & could be multiple reasons behind.
>
> We agree with the point and find that learning self-similarity is also effective in image classification. We validated it on CIFAR-10 and -100, which are the popular image classification benchmarks. For the experiments, we organize two basic blocks: a light-weighted version of SELFY block (SELFY$_l$) and a spatial convolutional block (SCB) with four 3x3 spatial convolutional layers as the counterpart. Both blocks have the same receptive field as ($9 \times 9$), and the similar FLOPs. We insert each of the basic blocks and their combinations (please refer to Fig.4c,4d, and 4f of our paper) into the middle of ResNet20.
> Table D summarizes the results on CIFAR-10 and -100. ResNet20, which is our baseline, performs as 8.44 \%p and 30.34 \%p at top-1 error on CIFAR-10 and -100, respectively.
> ResNet20 with SELFY$_l$ reduces the top-1 errors to 1.1. \%p and 1.98 \%p on the both benchmarks. The results are in alignment with those in Table 3 of our paper; SELFYNet with $l=\{0\}$ that learns spatial self-similarity improves the accuracy on both TSN and TSM backbones. Interestingly, ResNet20 with SEFLY$_l$ is more accurate than ResNet with SCB. Furthermore, in terms of the block combinations (4th - 6th rows), SELFY$_l$ + SCB and SELFY$_l$ + SELFY$_l$ show the lowest top-1 error rate as 64.7 \%p and 27.50 \%p on CIFAR-10 and -100, respectively. These results indicate that the spatial self-similarity features and the ordinary visual features are complementary to each other also on image classification, which is consistent with the results in Table 5 of our paper.
>
> Table D: **The effect of spatial self-similarity for image classification.**
>
> |model|FLOPS (G)|# params (K)|CIFAR-10 err-1|CIFAR-100 err-1|
> |:---|:---:|:---:|:---:|:---:|
> |baseline (ResNet20)|35.79|220|8.44|30.34|
> |$+$ SCB|50.41|277|7.81|28.77|
> |$+$ SELFY$_l$ |49.99|254|7.34|28.36|
> |$+$ SCB $+$ SCB |65.02|333|7.17|27.83|
> |$+$ SELFY$_l$ $+$ SELFY$_l$  |64.16|287|6.79|**27.50**|
> |$+$ SELFY$_l$ $+$ SCB |64.59|310|**6.47**|27.59|
>
> > [R3] Besides these video based action benchmarks, it should be interesting to see results on a depth enriched data set (i.e., RGB-D), as missing depth information is one of the limitations from video data. Ideally, we should see similar good performance if the proposed methodology is effective for representation learning.
>
> We also agree with the point, and thus, we’ve been conducting the experiments on the depth enriched dataset, e.g. NTU RGB+D, to validate our methods. We will leave the comment as soon as the experiments are finished.

---

> > ### Author Response · Authors · 2020-11-24
> > **Experimental results on a depth enriched dataset (NTU RGB+D 120)**
> >
> > > [R3] Besides these video based action benchmarks, it should be interesting to see results on a depth enriched data set (i.e., RGB-D), as missing depth information is one of the limitations from video data. Ideally, we should see similar good performance if the proposed methodology is effective for representation learning.
> >
> > As the reviewer suggested, we have also validated our method on NTU RGB+D 120 dataset [1] for action recognition in RGB-D videos as follows. We use TSN-R18 and TSM-R18 two-stream networks, which consist of RGB and depth stream networks [2], as our backbone and add a single SELFY block, of which (L,U,V) is (5,9,9), into each network stream after res_3 stage. We follow the cross-subject evaluation criteria and report the top-1 accuracy. Table E summarizes the results. The SELFY block improves the accuracy of depth-stream networks as well as rgb-stream networks for both TSN-R18 and TSM-R18. In terms of the two-stream accuracy, the SELFY block also shows consistent improvement on both TSN-R18 and TSM-R18.
> >
> > Table E: **Performance comparison with TSN-R18 on NTU RGB+D 120**. The top-1 accuracy is reported, following the cross-subject evaluation criteria.
> >
> > |model|rgb|depth|rgb+depth|
> > |:---|:---:|:---:|:---:|
> > |TSN-R18|64.2|70.3|76.2|
> > |SELFYNet-TSN-R18|83.8|83.4|89.2|
> > |TSM-R18|85.6|83.9|89.5|
> > |SELFYNet-TSM-R18|86.3|84.6|90.1|
> >
> > In the following, we provide experimental details. We adopt the segment-based frame sampling method and sample a clip of 8 frames as input.  For training, we train the networks from the scratch using 8 rgb (or depth) images. We use a cosine learning rate schedule with the first 10 epochs for warm-up, with setting the total epochs to 50 and the initial learning rate to 0.1. For testing, we sample a single center-cropped video clip. Following cross-subject evaluation criteria, we evaluate top-1 prediction scores of the rgb- and depth-stream, and then, we average the two scores to obtain the final accuracy of the two-stream networks. Other details are the same as those in Appendix A.
> >
> > [1] J. Liu, A. Shahroudy, M. Perez, G. Wang, L.Y. Duan, and A.C. Kot. “NTU RGB+D 120: A large scale benchmark for 3D human activity understanding.” TPAMI. 2019. \
> > [2] Y. Wang, Y. Xiao, F. Xiong, W. Jiang, Z. Cao, J. T. Zhou, and J. Yuan. “3DV: 3D dynamic voxel for action recognition in depth video.” CVPR. 2020.

---

> ### Author Response · Authors · 2020-11-19
> **Response to Reviewer 3 (1/2)**
>
> Thanks for appreciating our work and providing constructive comments. Below are our responses and we will reflect them in our final manuscript.
>
> > [R3] From Figure. 2 or Section 3.2.2, the final STSS representation Z is the combination of original input V and the STSS feature F(S(V)), so kind of unclear how much contribution is made from the original V in terms of the final recognition performance. Maybe this part is addressed in the experimental section somewhere.
>
> Following your comments, we evaluate SELFYNet purely based on STSS features to see how much the ordinary visual feature $\mathbf{V}$ contributes for the final prediction. That is, we pass the STSS features, $\mathrm{ReLU}(\mathbf{F}^\star \times_5 \mathbf{W}_\theta) $, into the downstream layers without the visual features $\mathbf{V}$ (Eq.9 in our main paper). For the simplicity of description, we denote the relational feature $\mathrm{ReLU}(\mathbf{F}^\star \times_5 \mathbf{W}_\theta)$ by $\mathbf{R}$ .
> Table A compares the results of using different cases of the output tensor $\mathbf{Z}$ ($\mathbf{Z}= \mathbf{V}$, $\mathbf{Z}=\mathbf{R}$, and $\mathbf{Z}=\mathbf{R}+\mathbf{V}$) on SS-V1. Interestingly, SELFYNet using only $\mathbf{R}$ achieves 45.5 \%p at top-1 accuracy, which is higher as 2.5 \%p than the baseline. As we add $\mathbf{V}$ to $\mathbf{R}$, we obtain an additional gain as 2.9 \%p. It indicates that the STSS features and the visual features are complementary to each other. We will add the experiment in our final manuscript.
>
> Table A: **The effect of fusing STSS with ordinary visual features.**
>
> |model|feature|top-1|top-5|
> |:---|:---:|:---:|:---:|
> |TSM-R18|$\mathbf{V}$|43.0|72.3|
> |SELFYNet|$\mathbf{R}$|45.5|75.9|
> |SELFYNet (ours)|$\mathbf{R}+\mathbf{V}$|**48.4**|**77.6**|
>
> > [R3] The proposed neural block, SELFY, includes 3 parts as self-similarity transformation, feature extraction, and feature integration, it seems none of them is original but the combination of these put into the end-of-end NN structure is new as claimed by authors. There are some additional experiment results presented in section 4.3 (for different types of similarity, and for different feature extraction), still, it seems unclear how important or sensitive for each step within the whole framework based on TSN ResNets and TSM ResNets, given the complexity here.
>
> Each of the three stages in our SELFY block plays a crucial role in learning fine-grained STSS features. The first stage, i.e., STSS transformation, determines the volume of the STSS $(L,U,V)$. To validate the importance and sensitivity of the first stage, we conducted experiments that compare the performance with different $(L,U,V)$. The results on SS-V1 are shown in Table B below. In most cases, the larger STSS volume results in better accuracy.
>
> During the second and the third stages, i.e., feature extraction and integration, the SELFY block learns to capture fine-grained structural patterns within the STSS tensor, resulting in the richer STSS representation. To validate the importance of the two stages, we conducted experiments that compare the performance of different combinations of feature extraction and integration methods. The results on SS-V1 are shown in Table C below. Compared to the baseline, we improve top-1 accuracy by 1.0 \%p, 2.9 \%p, and 3.7 \%p by changing the feature extraction method from soft-argmax to MLP to convolution, respectively. We also obtain additional gain as 0.5 \%p and 1.2 \%p at the top-1 accuracy by replacing the feature-integrating FC layer with MLP and with stacked convolutional layers, respectively. These results demonstrate that our design choice of using convolutions along $(U,V)$ and $(H,W)$ is the most effective in learning the geometry-aware STSS representation.
>
> Table B: **Performance comparison with different volumes of SELFY block.**
>
> |model|$(L,U,V)$|FLOPs (G)|top-1|top-5|
> |:---|:---:|---|---|---|
> |TSM-R18 |-|14.6|43.0|72.3|
> |SELFYNet|$(1,9,9)$|15.3|47.1|76.3|
> |SELFYNet|$(3,9,9)$|16.3|47.8|76.7|
> |SELFYNet|$(5,9,9)$|17.3|48.4|77.6|
> |SELFYNet|$(7,9,9)$|18.3|**48.6**|77.7|
> |SELFYNet|$(11,9,9)$|20.2|48.4|77.6|
> |SELFYNet|$(5,5,5)$|17.1|47.8|77.1|
> |SELFYNet|$(5,13,13)$|18.4|48.4|77.8|
> |SELFYNet|$(5,17,17)$|19.8|**48.6**|**78.3**|
>
> Table C: **Ablation results on different feature extraction and feature integration methods.** Smax denotes the soft-argmax operation. MLP layer consists of four fully connected (FC) layers. The $1 \times 1 \times 1$ convolutional layer in the feature integration stage is omitted from this table.
>
> |model|feature extraction|feature integration|FLOPs (G)|top-1|top-5|
> |:---|:---:|:---:|:---:|:---:|:---:|
> |TSM-R18|-|-|14.6|43.0|72.3|
> |SELFYNet|Smax|FC|14.8|44.0|73.2|
> |SELFYNet|MLP|FC|15.4|45.9|75.1|
> |SELFYNet|Conv|FC|16.1|46.7|75.8|
> |SELFYNet|Conv|MLP|16.3|47.2|75.9|
> |SELFYNet (ours)|Conv|Conv|17.3|**48.4**|**77.6**|

---

### Official Review · AnonReviewer2 · 2020-10-27
**Interesting model to integrate bi-directional self-similarity into spatial-temporal representations with empirical results showing gains in performance over SOTA on motion-centric data but some concerns remain over applicability to data that requires more appearance-centric representation learning.**

**Rating:** 6
**Confidence:** 4

**Review:**

The paper introduces SELFY, a neural module that learns spatio-temporal self-similarity across longer timescales in both directions to obtain visual features that provide consistent empirical gains on three action recognition datasets. Ablation studies show that modeling longer, bi-directional motion similarity can help handle motion blurs and (artificially induced) occlusion.

Strengths:
+ The paper is well written and shows strong empirical gains over prior SOTA.
+ The ablation experiments on the effect of temporal length and the different feature extraction methods are nicely done and show the effect of different design choices.
+ The experiments for testing the robustness of the learned representations is nice to see and helps highlight the strength of the proposed approach.

Concerns:
- From what I can see, the contribution is in the modeling of (motion) similarity in both directions, across a longer time scale. My concern is that all of the chosen datasets are very motion-centric where temporal relationships are more important and this makes it somewhat advantageous to the proposed model and does not really allow us to ascertain its applicability to datasets where appearance plays a bigger role such as Kinetics or HMDB-51. In fact, the results on SS-V1 and SS-V2 show some gains (~2-3%) compared to the very closely related MotionSqueeze Network (Kwon et al, ECCV 2020) which computes correlation (rather than similarity) between adjacent frames (t and t+1). MSNet did not see any significant advances in datasets (HMDB-51 and Kinetics) where appearance plays a greater role and I am not sure how this proposed approach would help in those conditions.
- I assume the function sim(.) from Equation 1 is cosine similarity since it was not explicitly mentioned anywhere. Is there any specific reason the correlation function was not used? What is the effect of using correlation in place of cosine similarity? It would interesting to see the effect of this since it would essentially allow us to see how modeling the longer temporal context would affect MS Net (the most closely related network) and help highlight the contribution of the proposed model beyond the obvious empirical gains.
- While the robustness experiments test out the occlusion setting by cutting out a rectangular patch of a single center-frame, that does not, IMHO, shows robustness to occlusion since the features from surrounding frames (on both sides) will help mitigate this artificial occlusion. In practice, the occlusion can have a larger effect since it would not zero out the appearance but will add some noise into the feature extraction and hence provide a stronger challenge. Experiments on HMDB51, which has strong camera motion and hence introduces natural occlusions, would be a stronger experiment to show the generalization of the model beyond datasets with a strong motion-centric bias.

---

> ### Author Response · Authors · 2020-11-19
> **Response to Reviewer 2**
>
> Thanks for appreciating our work and providing constructive comments. Below are our responses and we will reflect them in our final manuscript.
>
> > [R2] My concern is that all of the chosen datasets are very motion-centric where temporal relationships are more important and this makes it somewhat advantageous to the proposed model and does not really allow us to ascertain its applicability to datasets where appearance plays a bigger role such as Kinetics or HMDB-51.
>
> We are conducting the experiments on the appearance-oriented datasets. We will leave the comment as soon as the experiments are finished.
>
> > [R2] I assume the function sim(.) from Equation 1 is cosine similarity since it was not explicitly mentioned anywhere. Is there any specific reason the correlation function was not used? What is the effect of using correlation in place of cosine similarity?
>
> As pointed out, we use the cosine similarity as our default similarity function. We have explored the effect of different similarity functions on learning STSS, and found that the cosine similarity performs the best. Table A compares the performance on SS-V1 with different similarity functions: cosine similarity (ours), dot product [1], and embedded Gaussian [2]. The experimental details are the same as those in Appendix A, except that we reduce the channel dimension $C$  of appearance feature $\mathbf{V}$ to 32 for the less GPU memory consumption. The cosine similarity gives better accuracy, but not much, by 0.1 \%p and 0.2 \%p than the dot product and the embedded Gaussian at top-1 accuracy. We will clarify this point in our final manuscript.
>
> Table A: **Results of different similarity functions.**
>
> |model|\# channels|similarity function|top-1|top-5|
> |:---|:---:|:---:|:---:|:---:|
> |TSM-R18|-|-|43.0|72.3|
> |SELFYNet (ours)|32|cosine|**47.8**|**77.1**|
> |SELFYNet|32|dot product|47.7|76.6|
> |SELFYNet|32|embedded Gaussian|47.6|76.8|
>
> [1] P. Fischer, et al. “FlowNet: learning optical flow with convolutional networks.” ICCV. 2015.\
> [2] X. Wang, R. Girshick, A. Gupta, and K. He. “Non-local neural networks.” CVPR. 2018.
>
> > [R2] It would be interesting to see the effect of this since it would essentially allow us to see how modeling the longer temporal context would affect MS Net (the most closely related network) and help highlight the contribution of the proposed model beyond the obvious empirical gains.
>
> Below are the comparisons between SELFYNet and MSNet [3] to demonstrate the effectiveness of STSS as well as our feature extraction methods. For an apple-to-apple comparison, we apply the kernel soft-argmax and the max-pooling operations (*KS+CM* in [3]) to the feature extraction method by following their official github code. Please note that their official code also uses the cosine similarity as the same as ours. As we restrict the temporal offset $l$ of the SELFY variant to $\{ +1 \}$, it is equivalent to MS module of which *feature transformation layers* are the standard 2D conv layers. All experimental details are the same as those in Appendix A.
> Table B summarizes the results on SS-V1. KS+CM method achieves 46.1 \%p at top-1 accuracy. As we enlarge the temporal window $L$ to 5, we obtain an additional gain as 1.3 \%p. The learnable convolution layers improve top-1 accuracy by 1.0 \%p in both cases. We will add this experiment in our final manuscript.
>
> Table B: **Performance comparison with MSNet.**
>
> | model | feature extraction method | $(L,U,V)$ | top-1 | top-5 |
> |:---|:---:|:---:|---:|---:|
> |TSM-R18|-|-|43.0|72.3|
> |SELFYNet|KS+CM|$(1,9,9)$|46.1|75.3|
> |SELFYNet|KS+CM|$(5,9,9)$|47.4|76.8|
> |SELFYNet|conv|$(1,9,9)$|47.1|76.3|
> |SELFYNet|conv|$(5,9,9)$|**48.4**|**77.6**|
>
> [3] H. Kwon, M. Kim, S. Kwak, and M. Cho. “MotionSqueeze: neural motion feature learning for video understanding.” ECCV. 2020.
>
> > [R2] the occlusion can have a larger effect since it would not zero out the appearance but will add some noise into the feature extraction and hence provide a stronger challenge. Experiments on HMDB51, which has strong camera motion and hence introduces natural occlusions, would be a stronger experiment to show the generalization of the model beyond datasets with a strong motion-centric bias.
>
> We are conducting the experiments on HMDB51. We will leave the comment as soon as the experiments are finished.

---

> > ### Author Response · Authors · 2020-11-24
> > **Experiments on HMDB51 to study the robustness to natural occlusions**
> >
> > > [R2] While the robustness experiments test out the occlusion setting by cutting out a rectangular patch of a single center-frame, that does not, IMHO, shows robustness to occlusion since the features from surrounding frames (on both sides) will help mitigate this artificial occlusion. In practice, the occlusion can have a larger effect since it would not zero out the appearance but will add some noise into the feature extraction and hence provide a stronger challenge. Experiments on HMDB51, which has strong camera motion and hence introduces natural occlusions, would be a stronger experiment to show the generalization of the model beyond datasets with a strong motion-centric bias.
> >
> > As the reviewer suggested, we have experimented on HMDB51 to study the robustness to natural occlusions; we compare the results of TSM-R18 and SELFYNet variants like Fig. 5a, 5b of our manuscript. Table E summarizes the results on HMDB51. Note that in this empirical study we train our models from scratch as in 3D-R18 [10] and thus the overall performance in the table is low compared to the state of the arts. The results show that SELFYNets with the long temporal range ({-2,...,2} and {-3,...,3}) outperforms other models in top-1 and top-5 accuracy, being consistent with the results in Fig. 5a, 5b of our paper. This demonstrates that the proposed STSS representation with a sufficient volume of neighborhood helps video-processing models to be more robust to natural occlusions in HMDB51.
> >
> > Table E. **Results of SELFYNets with different ranges of $l$ on HMDB51.**
> >
> > |model|spatial size $(H,W)$| range of $l$ | top-1 | top-5 |
> > |:---|:---:|:---:|---:|---:|
> > |3D-R18[10]|$(112,112)$|-|17.1|-|
> > |TSM-R18|$(112,112)$|-|21.8|53.5|
> > |TSM-R18|$(224,224)$|-|27.3|59.8|
> > |SELFYNet|$(224,224)$|{0}|27.4|60.2|
> > |SELFYNet|$(224,224)$|{1}|28.5|62.3|
> > |SELFYNet|$(224,224)$|{-1,0,1}|30.9|64.7|
> > |SELFYNet|$(224,224)$|{-2,...,2}|33.4|66.9|
> > |SELFYNet|$(224,224)$|{-3,...,3}|34.3|67.1|
> >
> > In the following, we provide experimental details. We adopt the dense frame sampling method [1] and sample a clip of 16 frames as input.  For training, we use a cosine learning rate schedule with the first 10 epochs for warm-up, with setting the total epochs to 50 and the initial learning rate to 0.1. For testing, we sample 10 uniform clips per video and average the softmax scores for the final prediction. Other details are the same as those in Appendix A.
> >
> > [1] J. Carreira and A. Zisserman, “Quo vadis, action recognition? a new model and the kinetics dataset.” CVPR. 2017\
> > [2] H. Kuehne, H. Jhuang, E. Garrote, T. Poggio, and T. Serre. “HMDB: A Large Video Datasbase for Human Motion Recognition.” ICCV. 2011\
> > [3] J. Lin, C. Gan, and S. Han. “TSM: Temporal Shift Module for Efficient Video Understanding.” ICCV. 2019.\
> > [4] Y. Li, B. Ji, X. Shi, J. Zhang, B. Kang, and L. Wang. “ TEA: Temporal Excitation and Aggregation for Action Recognition.” CVPR. 2020.\
> > [5] H. Kwon, M. Kim, S. Kwak, and M. Cho. “MotionSqueeze: neural motion feature learning for video understanding.” ECCV. 2020.\
> > [6] C. Feichtenhofer, H. Fan, J. Malik, and K. He. “SlowFast Networks for Video Recognition.” ICCV. 2019.\
> > [7] H. Wang, D. Tran, L. Torrensani, and M. Feiszli. “Video modeling with correlation networks.” CVPR, 2020.\
> > [8] C. Yang, Y. Xu, J. Shi, B. Dai, B. Zhou. “Temporal Pyramid Network for Action Recognition.” CVPR. 2020.\
> > [9] X. Wang, R. Girshick, A. Gupta, and K. He. “Non-local neural networks.” CVPR. 2018.\
> > [10] K. Hara, H. Kataoka, and Y. Satoh. “Can Spatiotemporal 3D CNNs Retrace the History of 2D CNNs and ImageNet?.” CVPR. 2018.

---

> > > ### Comment · AnonReviewer2 · 2020-11-24
> > > **Thank you for the rather extensive evaluation and response**
> > >
> > > Thanks for providing the clarifications. I really appreciate your very detailed responses! Having read the other reviews and the author's responses, I feel that the paper makes a good contribution to learning good representations for visual understanding.  The additional experiments on HMDB51 and Kinetics show the generalization ability of the approach to beyond motion-centric video data. Overall, I think this is a good paper.

---

> > ### Author Response · Authors · 2020-11-24
> > **Experimental results on appearance-oriented datasets (Kinetics and HMDB51)**
> >
> > > [R2] My concern is that all of the chosen datasets are very motion-centric where temporal relationships are more important and this makes it somewhat advantageous to the proposed model and does not really allow us to ascertain its applicability to datasets where appearance plays a bigger role such as Kinetics or HMDB-51. In fact, the results on SS-V1 and SS-V2 show some gains (~2-3%) compared to the very closely related MotionSqueeze Network (Kwon et al, ECCV 2020) which computes correlation (rather than similarity) between adjacent frames (t and t+1). MSNet in datasets (HMDB-51 and Kinetics) where appearance plays a greater role and I am not sure how this proposed approach would help in those conditions.
> >
> > As you suggested, we have validated our method on the appearance-oriented datasets, Kinetics [1] and HMDB51 [2]. The results are compared to recent state-of-the-art methods below.
> >
> > Table C below summarizes the results on Kinetics. The first section of the table shows the results of the models with ResNet50 using 16 frames [3,4,5], which is the same size of backbone and input as ours. The second and third section of the table shows the results of the models with the same backbone (ResNet50) using 32 frames [6,7,8], and a larger backbone (ResNet101) using a bigger input [6,7,8], respectively. The proposed SELFYNet, the last section of the table, obtains a clear improvement of 2.4 \%p in top-1 accuracy compared to the TSM baseline [3], achieving the best performance among the models with ResNet50 using 16 frames.
> >
> > Table C. **Results of the state-of-the-art methods on Kinetics.**
> >
> > |model|backbone|# frames|top-1|top-5|
> > |:---|:---:|:---:|:---:|:---:|
> > |TSM [3]|ResNet 50|16|74.7|-|
> > |TEA [4]|ResNet 50|16|76.1|92.5|
> > |MSNet-TSM [5]|ResNet 50|16|76.4|-|
> > |------------------------------|---------------|----------|------|------|
> > |SlowFast 8$\times$8 [6] |ResNet 50|32|77.0|-|
> > |CorrNet [7]|ResNet 50|32|77.2|92.6|
> > |TPN [8]|ResNet 50|32|77.7|93.3|
> > |------------------------------|---------------|----------|------|------|
> > |SlowFast 16$\times$8 [6]|ResNet 101|32|77.9|93.2|
> > |SlowFast 16$\times$8 [6]|ResNet 101|64|78.9|93.5|
> > |TPN [8]|ResNet 101|32|78.9|93.9|
> > |CorrNet [7]|ResNet 101|32|79.2|-|
> > |------------------------------|---------------|----------|------|------|
> > |SELFYNet-TSM (ours)|ResNet 50|16|77.1|93.0|
> >
> > Table D below summarizes the results on HMDB51. We compare SELFYNet with the TSM baseline and MSNet [5]. SELFYNet obtains a significant improvement of 4.7 \%p in top-1 accuracy compared to the baseline, and outperforms MSNet. We observe that our method shows a bigger improvement on HMDB51 than Kinetics since HMDB51 clips have stronger motions as the reviewer mentioned.
> >
> > Table D. **Results of the proposed method on HMDB51.**
> >
> > |model|backbone|# frames|top-1|top-5|
> > |:---|:---:|:---:|:---:|:---:|
> > |TSM (our impl.)|ResNet 50|16|73.3|94.2|
> > |MSNet-TSM [5]|ResNet 50|16|77.4|-|
> > |SELFYNet-TSM (ours)|ResNet 50|16|78.0|95.6|
> >
> > The reviewer’s concern on our approach appears to come from that the recent motion-learning method of MSNet [5] does not show a significant gain on Kinetics. In our opinion, the limited gain is more related to the use of limited resources in training, e.g., in terms of the size of the backbone or the number of input frames, than to the use of correlations. As can be seen in Table C, SlowFast network [6] achieves better performance when a larger backbone or a longer input is used. Also, CorrNet [7] , which learns to extract motion features similarly to MSNet, obtains the state-of-the-art accuracy with ResNet101 backbone using 32 frames. While the results of SELFYNet with a larger backbone using longer input is not available for now due to the lack of time and GPU resources, we believe that SELFYNet can achieve state-of-the-art performance in the same conditions with aforementioned methods. We will do our best to add such large-scale experiments to our final version.
> >
> > In the following, we provide implementation details for our experiments. For both datasets, we adopt the dense frame sampling method [1] and sample a clip of 16 frames. For training on Kinetics, we use a cosine learning rate schedule with the first 10 epochs for warm-up. We set the initial learning rate to 0.01 and total epochs to 65. For training on HMDB51, we fine-tune the Kinetics pre-trained model like as recent approaches [1,3,4,5] and adopt a cosine learning rate schedule with 5 epochs for warm-up. We set the initial learning rate to 0.001 and total epochs to 35. For testing, we sample 10 uniform clips per video and average the softmax scores for the final prediction. We follow the strategy of non-local networks [9] to pre-process the frames and take 3 crops as input. Other experimental details are the same as those in Appendix A of our paper.

---

### Official Review · AnonReviewer4 · 2020-10-28
**Good paper with good results**

**Rating:** 6
**Confidence:** 4

**Review:**

#### General
This paper proposes spatio-temporal self-similarity (STSS), which captures structural patterns in space and time, for action recognition from videos.
Overall, I would like to recommend ICLR to accept the paper.
Pros and Cons I found in the paper are summarized as follows.


#### Pros.
1. SELFY, the proposed neural block that implements STSS, is a good extention and adjustment of prior self-similarity works to modern CNNs, resulted in achieving state-of-the-art performance on different datasets.
1. The ablation study reveals that SELFY with a long temporal range successfully incorporate the temporal information without explicitly exploiting optical flows.
1. The paper is generally well and clearly written.
1. The literature review is thorough, and the work is well contextualized in the literatures.

#### Cons
1. The novelty and the originality of the paper may be relatively low because the concept of the STSS itself was already proposed in prior studies as mentioned in the paper.
1. The reasoning or intuition behind some of the design choices are not always clear. For example,
    - The output of the feature extraction block is $F \in \mathbb{R}^{T \times X \times Y \times L \times C_F}$, and the authors say
        > The dimension of L is preserved to extract motion information across different temporal offsets in a consistent manner.

        But the motion information across different temporal offsets can be extracted in the feature extraction module if MLP or convolution is employed in the feature extraction block. In other words, even if the output is $F \in \mathbb{R}^{T \times X \times Y \times C_F}$, the motion information can be encoded in the output tensor. Possibly the authors empirically found the present design choice is better. If so, how does the performance change if $L$ is not preserved?
    - In the feature integration block, firstly $3 \times 3$ spatial convolution kernel is applied, then the temporal offset ($L$) and the channels ($C^*_F$) dimension is flattened, and finally $1 \times 1 \times 1$ spatio-temporal convolution is applied. Is this design chosen experimentally or is there any intuition behind?

#### Minor comments
1. It may be interesting to discuss the relationship with self-attention based networks such as [1-3]
1. About Table 7 in Appendix, how is the result if SELFY block is used in $res_1$ and $res_5$? I guess these option do not work well, but I think it is beneficial to list all the results.

[1] Hu+, Local Relation Networks for Image Recognition, ICCV 2019
[2] Ramachandran+, Stand-Alone Self-Attention in Vision Models, NeurIPS 2019
[3] Zhao+, Exploring Self-attention for Image Recognition, CVPR 2020

---

> ### Author Response · Authors · 2020-11-19
> **Response to Reviewer 4 (2/2)**
>
> > [R4] In the feature integration block, firstly 3×3 spatial convolution kernel is applied, then the temporal offset (L) and the channels (CF∗) dimension is flattened, and finally 1×1×1 spatio-temporal convolution is applied. Is this design chosen experimentally or is there any intuition behind?
>
> Our intuition is that each of $L$ channel-groups in the flattened $LC_{F^∗}$-dimensional STSS representation can be viewed as a spatial self-similarity feature within each relative temporal offset so that the $1 \times 1 \times 1$ convolutional layer, i.e. fully connected layer, is able to reduce the flattened feature to the $C$-dimensional feature vector considering the information of different temporal offsets. We also empirically found that a pooling operation such as max pooling or average pooling, which reduces $\mathbf{F} \in \mathbb{R}^{T \times X \times Y \times L \times C_{F^*}}$ along $L$ dimension, significantly degrades the performance. We conjecture that such pooling operations, which get rid of the information of temporal offsets, damage long-range temporal modeling.
>
> > [R4] It may be interesting to discuss the relationship with self-attention based networks such as [1-3].
>
> The local self-attention [6,7,8] and our method have a common denominator of using the self-similarity tensor but use it in a very different way and purpose. The local self-attention mechanism aims to aggregate the local context features using the self-similarity tensor and it thus uses the self-similarity values as attention weights for feature aggregation. However, our method aims to learn a generalized motion representation from the local STSS, so the final STSS representation is directly fed into the neural network instead of multiplying it to local context features.
> For a clear comparison, we conduct an ablation experiment as follows. We add a single *spatio-temporal* local self-attention layer, extended from [7], after $res_3$ followed by feature integration layers in Sec.3.2.2. All experimental details are the same as those in Appendix A, except that we reduce the channel dimension $C$  of appearance feature $\mathbf{V}$ to 32. Table C summarizes the results on SS-V1. The spatio-temporal local self-attention layer is accurate as 43.8 \%p at top-1 accuracy, and both of SELFY blocks using the embedded Gaussian and the cosine similarity outperform the local self-attention by achieving top-1 accuracy as 47.6 \%p and 47.8 \%p, respectively. These results are in alignment with the prior work [9], which reveals that the self-attention mechanism hardly captures motion features in video. We will add this to the discussion section in our final manuscript.
>
> Table C: **Performance comparison with the local self-attention.** R.P.E. is an abbreviation for relative positional embeddings.
>
> |model|similarity function|feature extraction method|top-1|top-5|
> |:---|:---:|:---:|:---:|:---:|
> |TSM-R18|-|-|43.0|72.3|
> |SELFYNet|embedded Gaussian|multiplication with $\mathbf{V}$+R.P.E|43.8|72.9|
> |SELFYNet|embedded Gaussian|conv|47.6|76.8|
> |SELFYNet (ours)|cosine|conv|**47.8**|**77.1**|
>
> [6] H. Hu, Z. Zhang, Z. Xie, and S. Lin. “Local relation networks for image recognition.” ICCV. 2019.\
> [7] P. Ramachandran, N. Parmar, A. Vaswani, I. Vello, A. Levskaya, and J. Shlens. “Stand-alone self-attention in vision models.” NIPS. 2019.\
> [8] H. Zhao, J. Jia, and V. Koltun. “Exploring self-attention for image recognition.” CVPR. 2020.\
> [9] X. Liu, J.Y. Lee, and H. Jin. “Learning video representations from correspondence proposals.” CVPR. 2019.
>
> > [R4] About Table 7 in Appendix, how is the result if SELFY block is used in res1  and res5 ? I guess these options do not work well, but I think it is beneficial to list all the results.
>
> We conducted the additional experiments and the results on SS-V1 are in Table D below, where we add the SELFY block to the different places. Here, we set the spatial resolution of $\mathbf{V}$ to $14 \times 14$. The SELFY block after $pool_1$ improves 2.7 \%p at top-1 accuracy, while the SELFY block after $res_5$ does not. We conjecture that the spatial resolution ($7 \times 7$) is too small to extract meaningful motion features for action recognition. We will update Table 7 in our manuscript.
>
> Table D: **Performance comparison with different positions of SELFY block.**
>
> |model|position|top-1|top-5|
> |:---|---|---|---|
> |TSM-R18|-|43.0|72.3|
> |SELFYNet |after $pool_1$|45.7|74.6|
> | SELFYNet|after $res_2$|47.2|76.6|
> |SELFYNet |after $res_3$|**48.4**|**77.6**|
> |SELFYNet |after $res_4$|46.6|76.0|
> |SELFYNet |after $res_5$|42.8|72.6|

---

> > ### Comment · AnonReviewer4 · 2020-11-21
> > **Thanks for the clarification**
> >
> > Thank you for the clarification and the effort for the additional experiments.
> > Taking the response together with other reviewers' comments and the authors' response to them, I am still positive in accepting the paper, and thus would like to keep my rating.
> > In my opinion, the clarified novelty still sounds moderate, but combined with the detailed experiment and the analysis, I think the paper is valuable to the community.
> > I recommend the authors to include the clarification of the novelty, intuitions behind the design choices, and the discussion on the relation to the local and global self-attention networks in the final version.

---

> ### Author Response · Authors · 2020-11-19
> **Response to Reviewer 4 (1/2)**
>
> Thanks for appreciating our work and providing constructive comments. Below are our responses and we will reflect them in our final manuscript.
>
> > [R4] The novelty and the originality of the paper may be relatively low because the concept of the STSS itself was already proposed in prior studies as mentioned in the paper.
>
> While the concept of STSS is already proposed in the prior work [1,2], there are important differences in the way of using STSS. The main contribution of our work is to use STSS as a basis for end-to-end learning a powerful long-term relational representation rather than using STSS feature itself as in the prior work. This is motivated by our novel view on the conventional motion (i.e., optical flow) as one-hot cross-similarity, which is a form of sparse and temporally-restricted STSS as briefly mentioned in our introduction. To this end, we explore different representation learning strategies (Sec.3.2.1 & 3.2.2) and propose our model that consists of convolutions along the (L,U,V) dimension (Sec.3.2.1), which learn relational patterns among similarity values, and spatial convolutions (Sec 3.2.2), which integrate pixel-wise STSS features by extending the receptive fields.
>
> Furthermore, unlike the previous methods, we propose to combine it with the ordinary visual feature (via addition in Eq.(9)) as a complementary one. This is also important. How complementary are they to each other? We find a clear synergic effect of combining STSS with visual features. Denoting the relational feature $\mathrm{ReLU}(\mathbf{F}^\star \times_5 \mathbf{W}_\theta)$ by $\mathbf{R}$ and the ordinary visual feature by $\mathbf{V}$, Table A below compares the cases of $\mathbf{V}$, $\mathbf{R}$, and $\mathbf{R}+\mathbf{V}$ on SS-V1. As we use the combination of $\mathbf{V}$ and $\mathbf{R}$, we obtain the additional gain of 2.9 \%p.
>
> Table A: **The effect of STSS representation ($\mathbf{R}$) with the ordinary visual feature ($\mathbf{V}$)**
>
> |model|feature|top-1|top-5|
> |:---|:---:|:---:|:---:|
> |TSM-R18|$\mathbf{V}$|43.0|72.3|
> |SELFYNet|$\mathbf{R}$|45.5|75.9|
> |SELFYNet (ours)|$\mathbf{R}$+$\mathbf{V}$|**48.4**|**77.6**|
>
> [1] E. Shechtman  and M. Irani. “Matching local self-similarities across images and videos.” CVPR, 2007.\
> [2] I.N Junejo, E. Dexter, I. Laptev, and P. Perez. “Cross-view action recognition from
> temporal self-similarities.” ECCV, 2008.
>
>
> > [R4] The reasoning or intuition behind some of the design choices are not always clear….  In other words, even if the output is F∈RT×X×Y×CF, the motion information can be encoded in the output tensor. Possibly the authors empirically found the present design choice is better. If so, how does the performance change if L is not preserved?
>
> The design choice of preserving the $L$ dimension to the last part of SELFY is inspired by the recent action recognition approach [3,4,5]; they maintain the temporal resolution throughout the architecture to the global pooling layer before classification. E.g., The work of [4] finds that holding fine temporal resolution is helpful for capturing detailed motion, and the architecture is thus designed to use no temporal downsampling layers. To validate our design choice, we have conducted experiments regarding the $L$ dimension: (1) preserving $L$ (ours), (2) reducing $L$ once via a convolution layer, and (3) reducing $L$ gradually via convolution layers. Table B below shows the results on SS-V1. Our method (1) shows the best performance at top-1 and top-5 accuracy. The case of (2) performs better than the case of (3), which implies that preserving L is more beneficial for capturing detailed information from STSS.
>
> Table B: **The effect of reducing the L in feature extraction.** $n * (k,k,k)$ denotes n convolution layers along the $(L,U,V)$ volume with a kernel size of $(k,k,k)$. We reduce the $L$ using convolutions without padding.
>
> |model|feature extraction method ($(L,U,V)=(5,9,9)$)|top-1|top-5|
> |:---|:---:|:---:|:---:|
> |SELFYNet (ours)|(1) $4*(1,3,3)$|**48.4**|**77.6**|
> |SELFYNet|(2) $4*(1,3,3) + 1*(5,1,1)$|47.5|76.8|
> |SELFYNet|(3) $2*(1,3,3)+1*(3,1,1)+2*(1,3,3)+1*(3,1,1)$|47.3|76.4|
>
> [3] J. Lin, C. Gan, and S. Han. “TSM: Temporal Shift Module for Efficient Video Understanding.” ICCV. 2019.\
> [4] C. Fiechtenhofer, H. Fan, J. Malik, and K. He. “SlowFast Networks for Video Recognition.” ICCV. 2019.\
> [5] C. Fiechtenhofer. “X3D: Expanding Architectures for Efficient Video Recognition.” CVPR. 2020.

---

### Author Response · Authors · 2020-11-24
**Paper revised**

We thank all reviewers for giving thoughtful comments and questions. The revised paper has been uploaded online. The major updates are summarized below:

- Add reasoning the model design choice in Sec. 3.2.
- Add a study of the effect of feature integration stage in Sec.4.3.
- Update new top-1 and top-5 accuracies of SELFYNet TSM-R18 (5,9,9): (48.2, 77.2) -> (48.4, 77.4) in Sec.4.3 and Appendix B.
- Add FLOPs in Sec.4.3 and Appendix B.
- Add a study of the effect of the original $\mathbf{V}$ in Appendix B.
- Update a study of the comparison with non-local methods in Appendix B.
- Add a study of the comparison with correlation-based methods in Appendix B.
- Add the results of $\textrm{pool}_1$ and $\textrm{res}_5$ in Appendix B.
- Add figures comparing our method with non-local methods and correlation-based methods in Appendix B (the figures are provided for better comprehension).
- Add a discussion section on the relationship to the local self-attention in Appendix C.

---

### Decision · Program_Chairs · 2021-01-07
**Final Decision**

**Decision:**

Reject

**Comment:**

This paper presents an approach to use spatio-temporal self-similarity (STSS) as a feature for a convolutional neural network for video understanding. The proposed approach extracts STSS as a descriptor capturing similarities between local spatio-temporal regions, and adds conventional layers such as soft-argmax, fully connected layers, and conv. layers on top of it.

On one hand, all of the reviewers agree that the novelty of the paper is limited. On the other hand, most of the reviewers (except R1) appreciated thoroughness of the experiments and ablations. In the end, the reviewers gave 3 marginally above the acceptance threshold ratings and 1 marginally below the threshold rating.

The AC views this paper as a borderline paper. None of the reviewers are excited about the paper, and it is a typical "Nice experiments with limited novelty" (by R1) paper. The concept of the STSS itself was already proposed in prior studies as mentioned in the paper and by the reviewers, and this paper 'engineers' a new way to take advantage of STSS without further theoratical or conceptual justifications on why it should work. The newly added Kinetics and HMDB results in the rebuttal are nice, but the impact of STSS seems to be minimal in these results.

Overall, the AC find the paper slighly lacking to be considered for ICLR.